# Ubiquitin pathway blockade reveals endogenous ADP-ribosylation marking PARP7 and AHR for degradation

Andrii Gorelik [ID][1][✉], Nina Đukić [ID][1], Rebecca Smith [ID][1], Chatrin Chatrin [ID][1], Osamu Suyari [ID][1], Jason Matthews[2,3] & Ivan Ahel [ID][1][✉]

## Abstract

**ADP-ribosylation is an important protein post-translational modification catalysed by a family of PARP enzymes in humans and is involved in DNA damage and immunity among other processes. While poly-ADP-ribosylation has been established as a protein degradation signal in several cases, the role of mono-ADP-ribosylation in protein turnover has remained elusive and mostly relies on overexpression systems. Here, we describe a way to visualise high levels of endogenous ADP-ribosylation by inhibiting the ubiquitin pathway. By blocking ubiquitylation/proteasome, we found that ADP-ribosylation by at least three different PARPs (PARP7, PARP1 and TNKS) can be greatly induced. We discovered that specific activation of the aryl hydrocarbon receptor (AHR) pathway in combination with the ubiquitin pathway inhibition promotes quantitative ADP-ribosylation of PARP7 targets, including the mono-ADP-ribosyltransferase PARP7 itself and AHR. We found that DTX2 is the E3 ligase responsible for degrading ADP-ribosylated PARP7, AHR and other PARP7 substrates. This PARP7-DTX2 crosstalk establishes a mechanism to rapidly shut down AHR-mediated transcription by decreasing its protein levels. Taken together, our findings uncover a paradigm where mono-ADP-ribosylation acts as a degradation mark.**

**Keywords** PARP7; Aryl Hydrocarbon Receptor; Ubiquitin; ADP-Ribose; Protein Degradation
**Subject Categories** Chromatin, Transcription & Genomics; Post-translational Modifications & Proteolysis

## Introduction

ADP-ribosylation (ADPr) of proteins participates in crucial processes linked to DNA damage and immune responses (Suskiewicz et al, 2023). This post-translational modification is primarily catalysed by a family of enzymes called PARPs that consists of 17 members in humans (Luscher et al, 2022). Different PARPs are triggered by various stimuli and then utilise donor substrate $NAD^+$ to attach ADP-ribose onto target protein substrates. While PARP1, PARP2 and tankyrases (TNKS1 and TNKS2) perform poly-ADPr (PARylation), which results in long chains of ADP-ribose units, the rest of the PARP family catalyses the addition of a single ADP-ribose moiety onto target amino acids resulting in mono-ADPr (MARylation) (Suskiewicz et al, 2023). ADPr is known to occur on a variety of amino acid acceptors such as serine, glutamate, aspartate, and cysteine (Suskiewicz et al, 2023; Vyas et al, 2014) and can be reversed by specific hydrolases, highlighting the dynamic nature of this modification (Rack et al, 2020).

The link between ADPr and the ubiquitin pathway has been studied for over a decade. The most well-known example of a PARP linked to protein degradation is TNKS. RNF146 is a poly-ADP-ribose-targeted E3 ubiquitin ligase that recognises TNKS substrates through its poly-ADP-ribose-binding WWE domain and targets them for proteasomal degradation through its ubiquitylation activity (DaRosa et al, 2015; Huang et al, 2009). Some of the examples are the degradation of axin, TNKS1 and TNKS2 (Huang et al, 2009).

Mono-ADPr has remained an elusive modification which is often difficult to detect at an endogenous level, and most studies have focused on overexpression systems (Longarini et al, 2023; Vyas et al, 2014).

Several mono-PARPs have been linked to protein degradation, including PARP7, but the mechanistic basis of this has been lacking (Kar et al, 2024; Zhang et al, 2020). Notably, PARP7 functions as part of a negative feedback loop regulating AHR signalling by acting as a regulator of the ligand-induced degradation of AHR. However, the mechanism of AHR degradation has remained poorly understood as previous studies looked at the indirect link through knockdown or knockout of PARP7, which led to increased AHR levels (Chen et al, 2025; MacPherson et al, 2013; Rijo et al, 2021). Whether this effect is transcriptional or post-translational has not been reported.

Here, we demonstrate that inhibiting the first enzyme in the ubiquitin cascade leads to a robust increase in ADPr targets. Activation of AHR further increases ADPr of PARP7 substrates. By

---

[1]Sir William Dunn School of Pathology, University of Oxford, Oxford OX1 3RE, UK. [2]Department of Nutrition, Institute of Basic Medical Sciences, University of Oslo, 0317 Oslo, Norway. [3]Department of Pharmacology and Toxicology, University of Toronto, Toronto, ON M5S 1A8, Canada. ✉E-mail: andrii.gorelik@path.ox.ac.uk; ivan.ahel@path.ox.ac.uk

screening for ADP-ribose-interacting E3 ligases, we identify DTX2 as the major E3 ligase responsible for the degradation of ADP-ribosylated PARP7 targets during AHR activation in a proteasome-dependent manner. We show that AHR is one of these targets, suggesting a previously unappreciated mechanism to restrict AHR-mediated transcription.

# Results

## Inhibition of the ubiquitin pathway induces ADPr by PARP7 and other PARPs

Several reports have suggested that ADPr can mark proteins for degradation, but detection and identification of the short-lived ADPr substrates have been elusive (Bhardwaj et al, 2017). We hypothesised that inhibiting the first enzyme (E1) in the ubiquitylation cascade with TAK243 or the proteasome with MG132 would reveal the ADPr substrates, which are rapidly turned over (Fig. 1A). Of note, PARP7 has been linked to protein degradation through its ADPr activity via unknown mechanisms (Zhang et al, 2020). As a model system, we chose the HCC44 lung cancer cell line, which we have shown previously is sensitive to PARP7 inhibition (Gorelik et al, 2025). Gratifyingly, upon E1 enzyme inhibition, we observed an evident appearance of previously undetectable ADP-ribosylated substrates using an ADP-ribose-specific antibody (Fig. 1B). In a titration experiment, a concentration of 1 μM of TAK243 was determined as the minimum concentration that induces ADPr signal (Appendix Fig. S1). We thus used this concentration in further experiments to avoid off-target effects and to prevent adverse effects on cell viability with higher TAK243 concentrations.

We then performed a time-course experiment with TAK243 and MG132 (Fig. 1C). Interestingly, the appearance of a ~200 kDa ADP-ribosylated substrate was apparent after just 1 h of TAK243 treatment, which we have previously suggested to relate to automodified PARP14 (Kar et al, 2024). At 2 h, two additional strong ADP-ribose signals appeared at 75 and 100 kDa, which peaked at 8 h of treatment. At 16 and 24 h of treatment, multiple other bands appeared, but the 75 and 100 kDa were suppressed. The MG132 treatment resulted in a similar, albeit weaker pattern (Fig. 1C).

Due to the possibility that TAK243 and MG132 can induce endoplasmic reticulum (ER) stress response (Hyer et al, 2018), we asked whether the ADPr signal arises from ER stress. We treated cells with ER stress inhibitors ISRIB and 4μ8c but saw no substantial reduction in ADPr signal (Appendix Fig. S2), suggesting that TAK243- and MG132-induced ADPr is not ER stress-dependent.

To test what PARPs result in this ADPr, we performed TAK243 treatment and used inhibitors against several widely expressed PARPs: olaparib for PARP1 and PARP2, AZ6102 for TNKS1 and TNKS2 (Johannes et al, 2015), RBN2397 for PARP7 (Gozgit et al, 2021) and RBN012759 for PARP14 (Schenkel et al, 2021) (Figs. 1A and 2A). Interestingly, the major band at 75 kDa was eliminated through RBN2397 (PARP7i) treatment (Fig. 2A). Other inhibitors did not visibly affect the ADPr signal despite the target engagement validated through the increased expression of TNKS and PARP14 (Bhardwaj et al, 2017; Kar et al, 2024). Due to the lack of reliable

commercial antibodies and the difficulty in detecting active PARP7 in HCC44 cells, we additionally used HiBiT-tagged PARP7 CT26 mouse colorectal carcinoma cells (Sanderson et al, 2023) (Fig. 2B). TAK243 treatment resulted in a similar ADPr pattern consistent with HCC44 cells and induced PARP7 levels as detected by the anti-HiBiT antibody. PARP7 inhibition with TAK243 treatment resulted in a downward HiBiT band shift, suggesting quantitative ADPr on PARP7 in TAK243 conditions without PARP7i. We made further use of the HiBiT tag to test whether PARP7 overlaps with the 75 kDa ADPr band and indeed observed overlap, further suggesting that the ADPr signal belongs to PARP7 automodification (Fig. 2B). We then performed a plate-based luminescence HiBiT assay using this CT26 cell line and confirmed a robust increase in PARP7 levels with TAK243, MG132 and RBN2397 (Fig. 2C).

As the treatment of HCC44 cells with TAK243 for 24 h resulted in substantial cell death, we did not test that time point in western blot experiments with this cell line. On the other hand, cells from another lung cancer cell line A549 were more tolerant towards the E1 inhibitor which allowed us to visualise higher levels of ADPr with a long-term treatment (Fig. 2D). Interestingly, in A549 cells, the E1 inhibitor-induced ADPr was sensitive to PARP7i (loss of a 75 kDa band), TNKSi (loss of 120–150 kDa bands) and PARP1/2i olaparib (loss of histone ADPr at ~20 kDa) (Palazzo et al, 2018) (Fig. 2D). Partial loss of signal was observed for the 200 kDa band in the PARP14i condition. This suggests that several PARPs and their substrates are turned over through ADPr marks.

We chose to perform follow-up experiments with a 4 h TAK243 treatment to minimise the negative effects on cell viability and to prevent cellular stress responses.

Overall, these data show that the inhibition of the ubiquitin-proteasome pathway allows visualisation of previously undetectable ADPr substrates.

## AHR activation increases PARP7-dependent ADPr

Intrigued by the strong PARP7-mediated ADP-ribose signal, we focused on the AHR signalling, which we and others have recently shown is important for PARP7 inhibitor response (Chen et al, 2025; Gorelik et al, 2025).

In the absence of ligand, cytosolic AHR is part of a multiprotein chaperone complex consisting of heat shock protein 90 (HSP90), aryl hydrocarbon receptor-interacting protein (AIP), c-Src and p23. After binding a ligand, AHR translocates to the nucleus, dissociates from the chaperone proteins and heterodimerises with AHR nuclear translocator (ARNT). The AHR/ARNT heterodimer complex binds to xenobiotic (dioxin) response elements (XRE/DRE) on DNA to promote transcription of target genes (Beischlag et al, 2008). AHR then undergoes rapid ubiquitin-dependent degradation by the proteasome in the cytoplasm, though the E3 ligase responsible for this degradation is unknown (Fig. 3A) (Davarinos and Pollenz, 1999; Polonio et al, 2025). We asked whether the transient ADPr of activated AHR could be visualised by blocking the ubiquitin pathway. Importantly, upon activation, the half-life of AHR decreases from 28 h to just 3 h (Ma and Baldwin, 2000). Consistent with the mechanism of AHR activation, its agonist tapinarof (Smith et al, 2017) led to decreased AHR levels. Interestingly, activation of AHR signalling with tapinarof, further increased the ADPr signal observed with E1 inhibition by TAK243

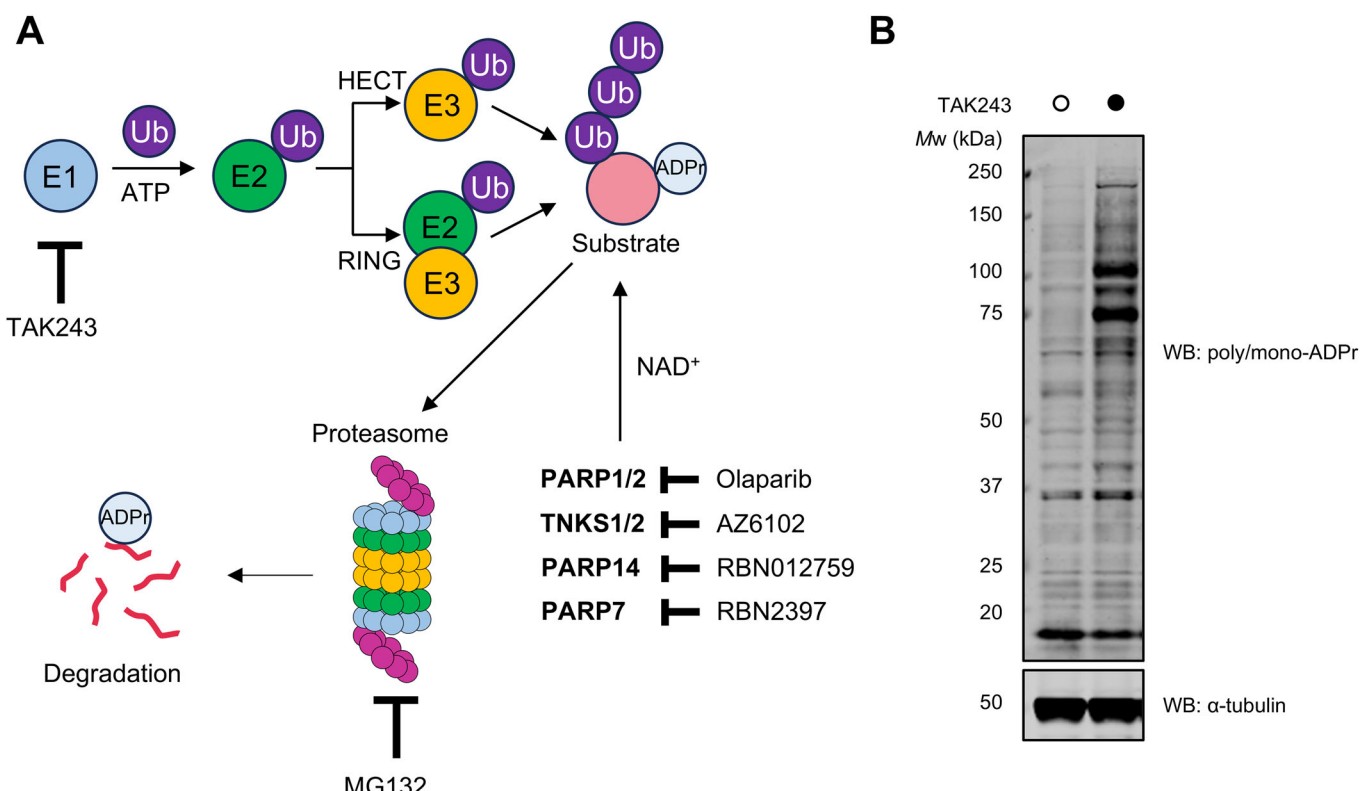

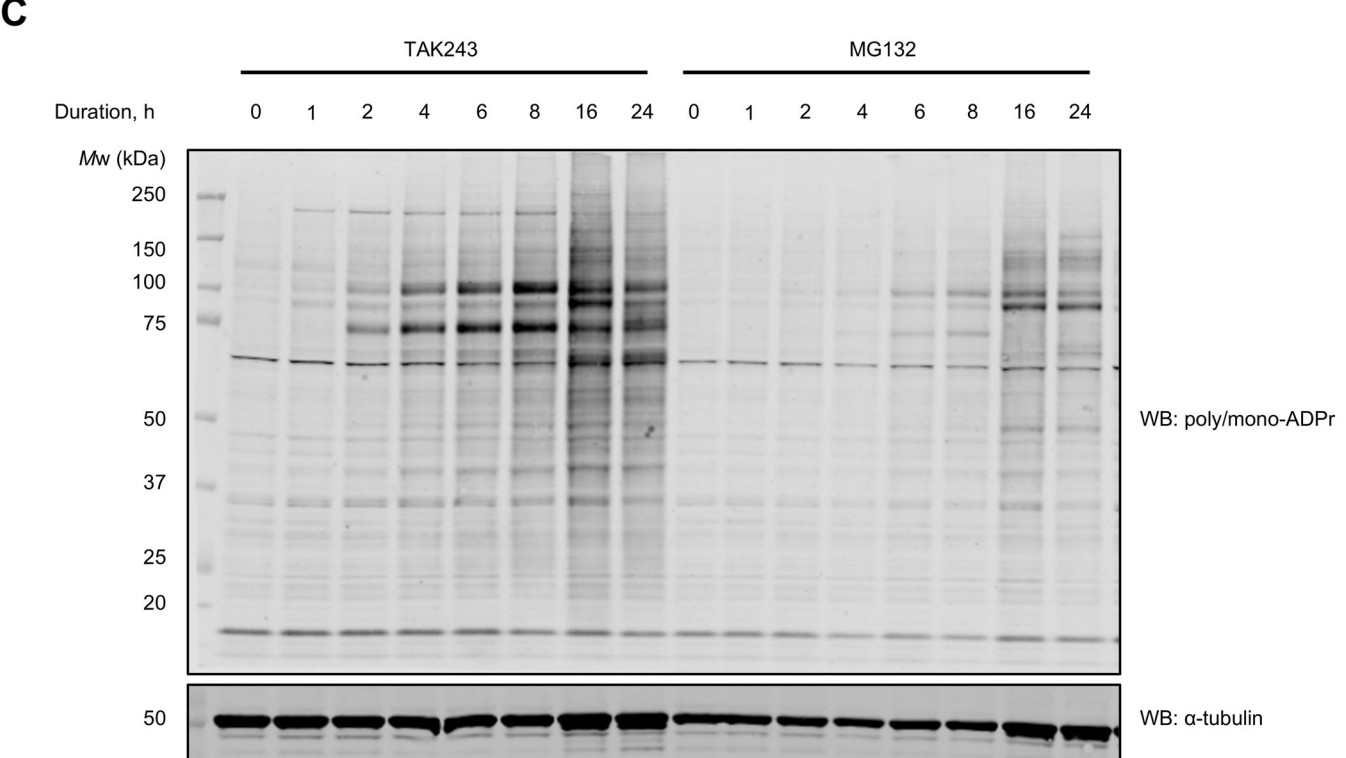

◄ **Figure 1. Inhibition of the E1 enzyme or the proteasome leads to increased ADPr.**

(A) Schematic overview of the ubiquitin-proteasome system, major PARPs and their inhibitors used in this study. Ub: ubiquitin, E1: ubiquitin-activating enzyme, E2: ubiquitin conjugating enzyme, E3: ubiquitin ligase (HECT or RING type). (B) HCC44 cells were treated with DMSO or 1 µM TAK243 for 4 h. (C) Time-course experiment with TAK243 and MG132 at 1 and 10 µM, respectively, in HCC44 cells. ADPr was assessed by western blotting using the anti-poly/mono-ADPr antibody, with α-tubulin as a loading control. Source data are available online for this figure.

in HCC44 cells (Fig. 3B). Additional ADPr targets appeared through this co-treatment which were completely removed with PARP7 inhibition. The MG132-tapinarof co-treatment also increased the 75 and 100 kDa bands (Fig. 3B). This suggests that AHR-regulated PARP7 substrates are transiently modified and then degraded. While PARP7 levels are kept very low, the PARP7 inhibitor treatment allowed its detection (Fig. 3B). Blotting for AHR revealed a band shift with TAK243-tapinarof and MG132-tapinarof co-treatments, which was prevented through PARP7 inhibition (Fig. 3B), suggesting that AHR is nearly quantitatively ADP-ribosylated under these conditions.

To further validate that the 100 kDa and 75 kDa ADPr substrates correspond to AHR and PARP7, respectively, we performed immunoprecipitation experiments. Endogenously HiBiT-tagged PARP7 was immunoprecipitated from HiBiT-PARP7 CT26 cells. Pulldown using the anti-HiBiT antibody revealed a single ADPr band at 75 kDa (Appendix Fig. S3A). Likewise, pulldown of AHR from HCC44 cells resulted in the enrichment of a 100 kDa band, corresponding to AHR (Appendix Fig. S3B). Thus, the 100 and 75 kDa ADPr substrates correspond to AHR and PARP7, respectively.

We next tested whether the AHR band shift is ADPr-dependent by performing ADPr hydrolase treatment on cell lysates from HCC44 cells (treated with tapinarof and TAK243). We used *Streptococcus pyogenes* macrodomain (*Spy*MacroD), which can hydrolase Glu/Asp/Cys-linked ADPr (Voorneveld et al, 2021) (the major PARP7-mediated ADPr linkages previously reported (Pala-valli Parsons et al, 2021; Rodriguez et al, 2021; Wierbilowicz et al, 2025)). Gratifyingly, the tapinarof-TAK243-induced ADPr signal was AHR band shift collapsed with wild type but not inactive (C119T mutant) *Spy*MacroD, while the arginine-ADPr specific hydrolase ARH1 had no effect on AHR (Appendix Fig. S4).

To test whether the AHR activation can boost degradation-targeted PARP7-mediated ADPr in a different cell line, we used wild type and PARP7-deficient breast cancer MCF7 cells generated previously (Rasmussen et al, 2021). Reproducibly, the TAK243-tapinarof combination increased the ADPr signal, which was removed through PARP7 inhibition (Fig. 3C). These bands were not observed in the PARP7 knockout MCF7 cells, suggesting that the ADPr events are PARP7-specific. Again, upon TAK243-tapinarof treatment, AHR exhibited a shift in molecular weight reversed by PARP7i and suggestive of quantitative ADPr (Fig. 3C).

In summary, our data indicate that PARP7 ADPr targets are induced through AHR activation and undergo degradation.

## DTX2 E3 ubiquitin ligase targets PARP7 ADPr substrates for degradation

Next, we hypothesised that E3 ligases that have ADPr recognition domains (DTC and WWE) may target ADP-ribosylated proteins for degradation. Among these are Deltex E3 ligases, RNF114 (as

well as related RNF125, RNF138 and RNF166), RNF146 and HUWE1 (Kloet et al, 2025; Perrard et al, 2025; Zhang et al, 2020; Zhang et al, 2011; Zhu et al, 2022). To determine which E3 ubiquitin ligase recognises and targets ADP-ribosylated proteins for degradation we performed a screen with a panel of siRNAs against all E3 ubiquitin ligases from the Deltex family (DTX1, DTX2, DTX3, DTX3L and DTX4), as well as RNF114, RNF125, RNF138, RNF166, HUWE1, and RNF146 in HCC44 cells (Fig. 4A; Appendix Fig. S5). In the absence of treatment, none of the E3 ligases substantially increased the ADP-ribose signal apart from RNF146, which served as a positive control for TNKS1 and TNKS2 ADPr modification as RNF146 specifically controls PARylation-dependent degradation of TNKS (Bhardwaj et al, 2017). Strikingly, the addition of tapinarof to activate AHR revealed several ADPr substrates exclusively upon DTX2 knockdown. The pattern was remarkably similar to that observed with TAK243 and TAK243-tapinarof treatments (Figs. 1–3). We observed two major bands at 75 and 100 kDa, plus additional ADPr bands at ~120 and 65 kDa. This result was reproduced using an additional siRNA sequence (Fig. 4B). To test whether the DTX2 and RNF146 knockdowns produce ADPr mediated by PARP7 and TNKS, respectively, we performed additional inhibitor treatments. Treatment of DTX2 knockdown HCC44 cells with PARP7i removed the tapinarof-induced ADPr signal, while the ADPr signal induced by RNF146 knockdown was removed upon TNKS inhibition (Appendix Fig. S6). This suggests that ADPr induced by DTX2 and RNF146 is catalysed by PARP7 and TNKS, respectively.

To investigate whether this phenomenon extends to other cell lines, we performed a DTX2 knockdown in A549 cells and observed the appearance of 75 and 100 kDa bands upon tapinarof treatment (Fig. 4C). Interestingly, AHR levels were increased in DTX2 knockdown cells already at the basal level, further pointing out the role of DTX in the regulation of AHR levels even under normal growth conditions (Fig. 4B,C). Western blotting for AHR revealed a partial overlap with the 100 kDa ADPr band, suggesting that the ADPr signal comes from the slower migrating ADP-ribosylated fraction of AHR (Fig. 4B,C).

We then generated a CRISPR/Cas9 knockout of DTX2 in A549 cells. Treatment of wild type cells with TAK243 resulted in an increased ADP-ribose signal, which was not the case when AHR was activated with tapinarof treatment (Fig. 5A). DTX2 knockout A549 cells showed an increase in ADPr with tapinarof alone (Fig. 5A). The pattern of this ADPr was very similar to the one observed with TAK243 treatment of wild-type A549 cells (Fig. 5A). This further confirmed our previous observations with a DTX2 knockdown in HCC44 and A549 cells (Fig. 4). This signal was PARP7-dependent as PARP7i eliminated the ADPr bands. Again, basal AHR levels were higher in DTX2 knockout cells, suggesting stabilisation (Fig. 5A). However, we noticed that tapinarof still resulted in a substantial reduction of AHR levels even in the DTX2 knockout. We hypothesised that AHR activation through 24 h

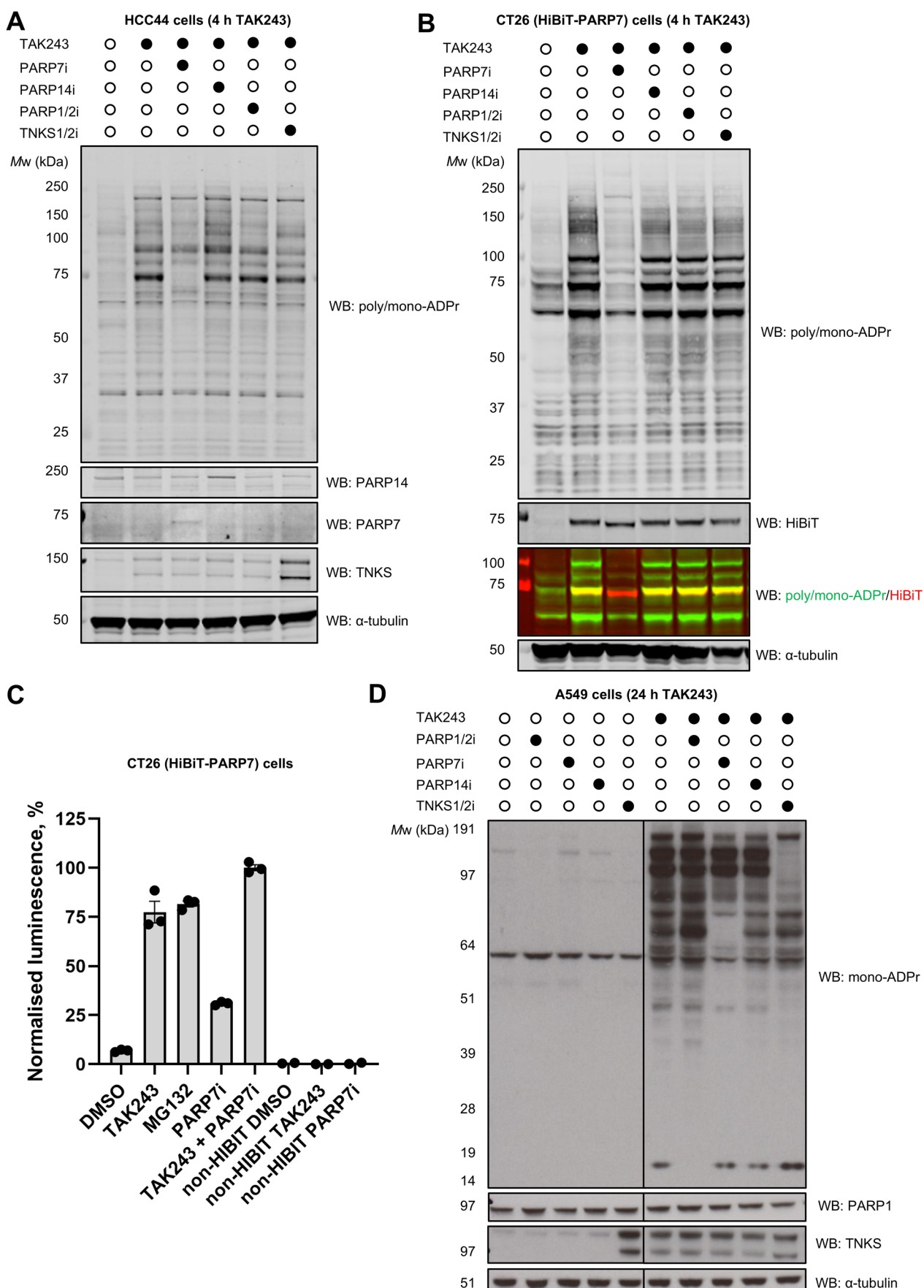

**Figure 2. PARP7 is the major PARP responsible for ADPr induced via short-term TAK243 treatment.**

(A) HCC44 cells pre-treated with DMSO or PARP inhibitors for 20 h were further treated with 1 μM TAK243 for 4 h. ADPr was assessed by western blotting using the anti-poly/mono-ADPr antibody, with α-tubulin as a loading control. (B) HiBiT-PARP7 CT26 cells pre-treated with DMSO or PARP inhibitors for 20 h were further treated with 1 μM TAK243 for 4 h. ADPr was assessed by western blotting using the anti-poly/mono-ADPr antibody, with α-tubulin as a loading control. Overlap of the signals was obtained using fluorescently labelled anti-rabbit (poly/mono-ADPr, green) and anti-mouse (HiBiT, red) secondary antibodies. (C) HiBiT luminescence assay in HiBiT-PARP7 CT26 cells with 1 μM TAK243, 10 μM MG132, 100 nM RBN2397 (PARP7i) treatments for 4 h showing induction of PARP7 levels (HiBiT luminescence). Wild-type CT26 cells (non-HiBiT) were used as a negative control. Data are shown as mean ± s.e.m. of $n = 3$ technical replicates. (D) A549 cells were co-treated with PARP inhibitors and 1 μM TAK243 for 24 h. PARP inhibitor concentrations in panels (A–C) are specified in the methods section. ADPr was assessed by western blotting using the anti-mono-ADPr antibody. α-tubulin was used as a loading control. The following PARP inhibitors were used: PARP1/2 (1 μM olaparib), PARP7 (100 nM RBN2397), PARP14 (RBN012759, 200 nM in Fig. 2A, B and 500 nM in Fig. 2D), TNKS (1 μM AZ6102). PARP14, PARP7, TNKS and PARP1 were detected with corresponding antibodies in relevant panels. Source data are available online for this figure.

tapinarof treatment could result in AHR degradation by E3 ubiquitin ligases other than DTX2. To rectify this, we performed tapinarof treatment for 2 h. Indeed, the shorter treatment resulted in noticeably more AHR in DTX2 knockout cells compared to wild-type cells (Fig. 5B).

Next, we wanted to investigate the functional effect of DTX2 knockout on AHR-mediated transcription. To address this, we performed real-time quantitative polymerase chain reaction (RT-qPCR) for major AHR target genes CYP1A1 and CYP1B1 (MacPherson et al, 2013) in wild type and DTX2 KO cells treated with tapinarof alone or in combination with PARP7i. AHR activation with tapinarof increased CYP1A1 and CYP1B1 levels, which was further elevated via the PARP7i-tapinarof combination (Fig. 5C,D). Strikingly, induction of CYP1A1 and CYP1B1 was substantially higher in DTX2 KO cells compared to wild-type cells (approximately sixfold for CYP1A1 and threefold for CYP1B1).

DTX2 has recently been shown to catalyse ADPr-ubiquitin hybrid modification (Chatrin et al, 2025; Zhu et al, 2022). We asked whether this could be the type of non-canonical ubiquitylation on AHR. To address this, we immunoprecipitated AHR and performed E1/E2/DTX2 reactions. The reaction resulted in a shift of the ADP-ribosylated AHR fraction and was sensitive to SpyMacroD (complete removal of ADPr signal), $NH_2OH$ (to hydrolyse the ester bond between ubiquitin and ADPr) and pan-specific deubiquitinase USP2 treatment (Appendix Fig. S7A). The latter two treatments collapsed the shifted bands. This suggests that DTX2 can modify ADPr with ubiquitin. We thought this could explain why the MG132-induced ADPr signal is lower than with TAK243 (ubiquitin masking ADPr recognition in the former case). However, when cell lysates were treated with USP2, the ADPr signal was unaffected (Appendix Fig. S7B). This suggests that while DTX2 can catalyse ADPr-ubiquitin on AHR, this modification is labile and/or is quickly turned over by various cellular hydrolases. Moreover, DTX2 weakly interacted with the ADP-ribosylated AHR in an immunoprecipitation experiment with no interaction observed upon PARP7i, suggesting a transient AHR-DTX2 complex and rapid AHR targeting to the proteasome (Appendix Fig. S8).

Thus, DTX2 is the major E3 ligase that targets PARP7 substrates for degradation upon AHR activation.

## Major PARP7-DTX2 degradation substrates are enriched in the nucleus

Unlike, other DELTEX E3 ligases, DTX2 is mainly localised in the nucleus (Ahmed et al, 2020). To investigate the localisation of

PARP7 ADPr substrates targeted for degradation, we performed immunofluorescence experiments in HCC44 cells (Fig. 6A). Upon TAK243 treatment, ADPr signal was localised throughout the cell with the vast majority of the signal in the nucleus (Fig. 6B). The combination of TAK243 with tapinarof, further enhanced ADPr in the nucleus (Fig. 6B). This ADPr increase was in agreement with the western blot data (Fig. 3B). PARP7 inhibition substantially reduced the ADP-ribose signal specifically in the nucleus (Fig. 6A,B). Consistent with AHR degradation through activation, tapinarof reduced AHR levels, while TAK243 treatment stabilised AHR (Fig. 6A,C). Treatment of cells with TAK243 resulted in decreased nuclear ubiquitylation (Appendix Fig. S9).

Next, we tested A549 cells under the same treatments and again observed accumulation of nuclear ADPr (Fig. 6D,E). The combination of tapinarof and TAK243 resulted in a small but significant increase in nuclear ADPr (Fig. 6D,E). This boost in ADPr could be reversed by PARP7i (Fig. 6D,E).

We then asked whether AHR activation in the absence of DTX2 could reveal localisation of PARP7 ADPr substrates (and thus DTX2 ubiquitylation substrates). Indeed, in DTX2 knockout A549 cells treated with tapinarof alone ADPr signal was increased, and this was reversed by PARP7 inhibition (Fig. 6D,E). Consistent with our western blot data, tapinarof treatment did not substantially affect ADPr signal in wild-type A549 cells (Fig. 6E), indicating the central role of DTX2 in the degradation of AHR-induced ADPr substrates. Akin to the result with HCC44 cells, tapinarof reduced AHR levels, while TAK243 treatment stabilised AHR (Fig. 6D,F).

To test the effects of TAK243 treatment on PARP7 levels and localisation, we used HiBiT-PARP7 CT26 cells. Both ADPr and PARP7 signals in the nucleus (detected via the anti-HiBiT antibody) were substantially induced through TAK243 treatment (Fig. 6G–I). PARP7 induction was in agreement with our western blot and luciferase assay results (Fig. 2B,C). HiBiT signal was not observed in wild-type CT26 cells, which served as a negative control (Appendix Fig. S9).

Taken together, our findings indicate that the majority of TAK243-induced ADPr substrates are enriched in the nucleus, which is consistent with the nuclear localisation and function of PARP7, AHR and DTX2.

## Discussion

The interplay of ADPr and ubiquitin has been previously noted, and our data show that it is even more widespread than previously thought. For instance, TRIP12 has been suggested to mark PARP1

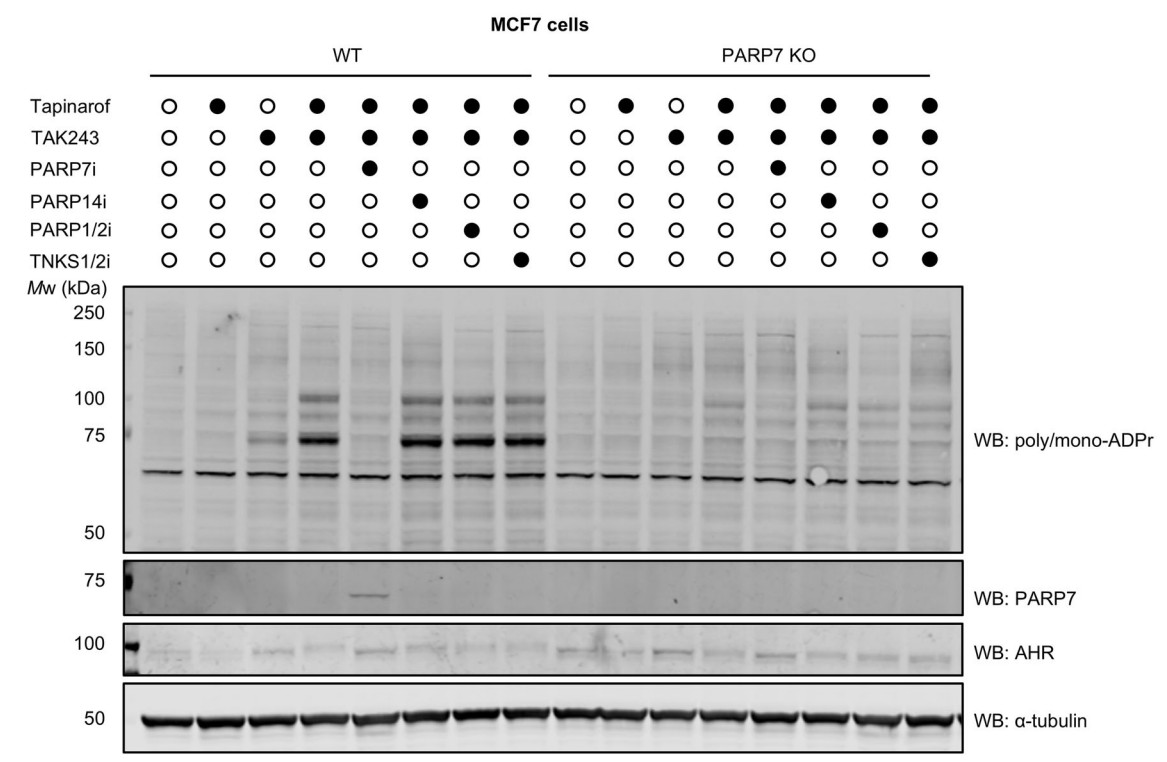

**A**

Tapinarof

*Cell Membrane*

*Nucleus*

Genes related to metabolism, immunity, toxicity, incl. **PARP7**

XRE/DRE

Proteasomal degradation of AHR

*Cytoplasm*

**B**

HCC44 cells

| Tapinarof | ○ | ● | ○ | ● | ● | ● | ● | ● | ○ | ● | ● | ● | ● | ● |
| TAK243 | ○ | ○ | ● | ● | ● | ● | ● | ● | ○ | ○ | ○ | ○ | ○ | ○ |
| MG132 | ○ | ○ | ○ | ○ | ○ | ○ | ○ | ○ | ● | ● | ● | ● | ● | ● |
| PARP7i | ○ | ○ | ○ | ○ | ● | ○ | ○ | ○ | ○ | ○ | ● | ○ | ○ | ○ |
| PARP14i | ○ | ○ | ○ | ○ | ○ | ● | ○ | ○ | ○ | ○ | ○ | ● | ○ | ○ |
| PARP1/2i | ○ | ○ | ○ | ○ | ○ | ○ | ● | ○ | ○ | ○ | ○ | ○ | ● | ○ |
| TNKS1/2i | ○ | ○ | ○ | ○ | ○ | ○ | ○ | ● | ○ | ○ | ○ | ○ | ○ | ● |

*M*w (kDa)

250
150
100
75

50

37

WB: poly/mono-ADPr

250 — WB: PARP14

75 — WB: PARP7

150 — WB: TNKS

100 — WB: AHR

50 — WB: α-tubulin

**C**

MCF7 cells

| | WT | | | | | | | | PARP7 KO | | | | | | | |
| Tapinarof | ○ | ● | ○ | ● | ● | ● | ● | ● | ○ | ● | ○ | ● | ● | ● | ● | ● |
| TAK243 | ○ | ○ | ● | ● | ● | ● | ● | ● | ○ | ○ | ● | ● | ● | ● | ● | ● |
| PARP7i | ○ | ○ | ○ | ○ | ● | ○ | ○ | ○ | ○ | ○ | ○ | ○ | ● | ○ | ○ | ○ |
| PARP14i | ○ | ○ | ○ | ○ | ○ | ● | ○ | ○ | ○ | ○ | ○ | ○ | ○ | ● | ○ | ○ |
| PARP1/2i | ○ | ○ | ○ | ○ | ○ | ○ | ● | ○ | ○ | ○ | ○ | ○ | ○ | ○ | ● | ○ |
| TNKS1/2i | ○ | ○ | ○ | ○ | ○ | ○ | ○ | ● | ○ | ○ | ○ | ○ | ○ | ○ | ○ | ● |

*M*w (kDa)
250
150
100
75

50

WB: poly/mono-ADPr

75 — WB: PARP7

100 — WB: AHR

50 — WB: α-tubulin

**Figure 3.  AHR activation and TAK243 treatment increase PARP7-dependent ADPr.**

(A) Schematic representation of AHR activation and degradation. Tapinarof is an agonist of AHR. (B) HCC44 cells were pre-treated with DMSO, 1 μM tapinarof and PARP inhibitors (PARP1/2 (1 μM olaparib), PARP7 (100 nM RBN2397), PARP14 (200 nM RBN012759), TNKS (1 μM AZ6102)) for 20 h followed by an additional 4 h of 1 μM TAK243 or 10 μM MG132 treatment. (C) MCF7 wild type (WT) and PARP7 knockout (KO) cells were treated as in (B). ADPr was assessed by western blotting using the anti-poly/mono-ADPr antibody. AHR, PARP7, TNKS and PARP14 were detected using corresponding antibodies with α-tubulin as a loading control. Source data are available online for this figure.

for proteasomal degradation (Gatti et al, 2020). On the other hand, an unusual ADPr-ubiquitin dual modification (Zhu et al, 2024; Zhu et al, 2022) was suggested to stabilise proteins (Perrard et al, 2025). While RNF146 has been linked to the degradation of poly-ADP-ribosylated TNKS substrates (DaRosa et al, 2015; Zhang et al, 2011), we propose that DTX2 performs an analogous function together with the mono-ADP-ribosyltransferase PARP7. Our data suggest that the two major degraded substrates are PARP7 itself and the transcription factor AHR.

There are several intriguing possibilities about the nature of ubiquitin linkages on AHR. For example, DTX2 may bind ADPr on AHR and PARP7 and perform canonical ubiquitylation on neighbouring lysines. Alternatively, PARP7 could produce the dual modification ADPr-Ub/MARUbylation (Chatrin et al, 2025; Lacoursiere et al, 2025; Wierbilowicz et al, 2025). While the ADPr-ubiquitin hybrid is an unstable modification (Zhu et al, 2022), our data suggest that DTX2 can also perform this PTM on AHR. New methods for the detection of ADPr-ubiquitin would help visualise this labile PTM in future studies. Finally, it is conceivable that both canonical and non-canonical ubiquitylation of AHR is a plausible outcome.

PARP7 is a protein which is expressed at very low levels in cells (Kamata et al, 2021). It appears that it is kept low through its own activity and degradation. Blocking PARP7 catalytic activity increases its levels (Gorelik et al, 2025; Yang et al, 2023; Zhang et al, 2020). Interestingly, the half-life of PARP7 has been reported to be only 4.5 min (Kamata et al, 2021). This is likely due to the very specialised role in AHR signalling where PARP7 is a transcriptional target of AHR and is a labile factor needed to control AHR-mediated transcription (Gorelik et al, 2025). Endogenous PARP7 modification has been difficult to demonstrate, and only a handful of studies overexpressed PARP7 or its substrates (Gomez et al, 2018; Rodriguez et al, 2021). Our data indicate that due to the degradation by DTX2, PARP7 substrates (including ADP-ribosylated PARP7 itself) are rapidly turned over.

We have previously shown that PARP7 inhibition and AHR activation cause extensive remodelling of the AHR-driven proteome, and this is concurrent with AHR stabilisation via PARP7i (Gorelik et al, 2025). Combined PARP7 inhibition and AHR activation leads to increased cancer cell death compared to the PARP7 inhibitor alone (Chen et al, 2025; Gorelik et al, 2025). While the involvement of PARP7 in AHR degradation has been proposed previously, the exact mechanism and which ubiquitin E3 ligases are involved have not been reported (Rijo et al, 2021). CUL4B E3 ligase depletion was reported to only partially stabilise AHR while PARP7 loss resulted in complete AHR stabilisation (Rijo et al, 2021). The ubiquitin pathway inhibition allowed us to uncover the elusive mechanism of degradation of AHR (Pollenz, 2002). Thus, PARP7 acts to quickly shut down AHR-mediated transcription via the E3 ligase DTX2. This is an intriguing insight into the complexity of

transcriptional and post-transcriptional/post-translational crosstalk between these proteins and supports the synthetic lethality observed by inhibiting PARP7 and activating AHR (Chen et al, 2022; Chen et al, 2025; Gorelik et al, 2025; MacPherson et al, 2013).

In our experiments, we utilised TAK243 to induce ADPr. However, long before this E1 inhibitor was discovered, the ubiquitin-dependent turnover of AHR was reported via the use of a temperature-sensitive E1 mutant that stabilised AHR upon TCDD (2,3,7,8-tetrachlorodibenzo-p-dioxin) treatment (Ma and Baldwin, 2000).

The fact that PARP7 is localised in the nucleus suggests that in the absence of nuclear localisation (without a ligand), cytoplasmic AHR does not encounter PARP7 and therefore does not undergo rapid degradation through the nuclear E3 ligase DTX2.

Degradation is not the only mechanism to shut down AHR. AHR repressor (AHRR) acts to suppress AHR-mediated transcription and is an AHR target itself that is expressed upon AHR activation (Fig. 3A) (Polonio et al, 2025). Therefore, it is possible that in some contexts AHRR mechanisms may be predominant as opposed to PARP7-DTX2-mediated degradation of AHR, and this may at least in part explain why certain cancer cell lines are resistant to PARP7 inhibition (Gozgit et al, 2021).

A similar role for PARP7 has been suggested recently for androgen receptor (AR) ADPr by the Paschal group. DTX2 can also recognise ADP-ribosylated AR and ubiquitylate it, causing AR degradation (Wierbilowicz et al, 2025). Interestingly, AHR can interact with CUL4B to form an E3 ubiquitin ligase complex and degrade androgen and oestrogen receptors (Ohtake et al, 2007). Further investigation is needed to reveal the role of the PARP7-DTX2 crosstalk in the turnover of these hormone receptors by AHR.

Future studies should make use of TAK243 to determine what other PARPs mediate the rapid turnover of their substrates, providing further mechanistic insights into this therapeutically important enzyme family. For example, PARP14 degradation through ADPr has been suggested (Dukic et al, 2023; Kar et al, 2024). In our experiments, we observed a band of ~200 kDa, which corresponds to the molecular weight of PARP14 but was not removed through PARP14 inhibition. Whether this is PARP14 modified by another PARP, modification on PARP4 (which has a similar molecular weight), or another substrate, remains to be established.

The question remains about the degradation of the unmodified PARP7 fraction. A previous study suggested that inactivation of PARP7 leads to its stabilisation. However, the half-life of inactive PARP7 is still very short at 30 min (Kamata et al, 2021). E3 ligases that control the degradation of inactive/unmodified PARP7 remain to be identified.

Interestingly, PARP7 inhibition appeared to partially rescue ubiquitin levels in TAK243-treated samples (Appendix Fig. S9),

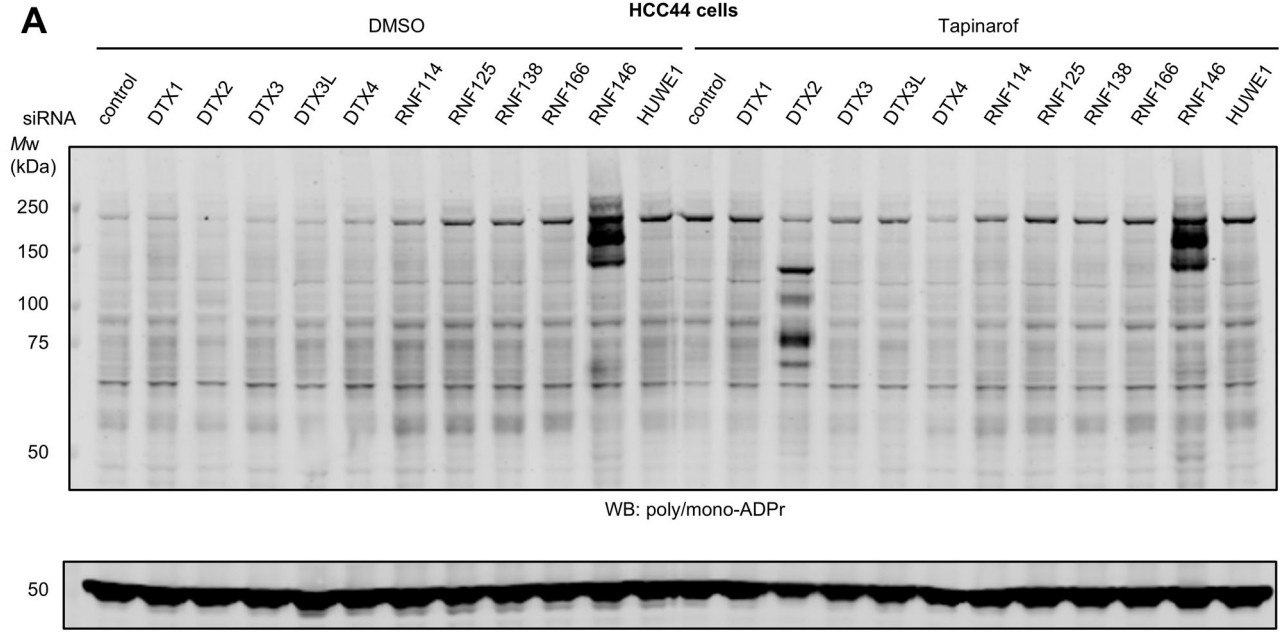

**A**

HCC44 cells

DMSO · Tapinarof

siRNA: control, DTX1, DTX2, DTX3, DTX3L, DTX4, RNF114, RNF125, RNF138, RNF166, RNF146, HUWE1

WB: poly/mono-ADPr

WB: α-tubulin

**B** HCC44 cells

DMSO · Tapinarof

siRNA: control, DTX2 #1, DTX2 #2, control, DTX2 #1, DTX2 #2

WB: poly/mono-ADPr

WB: AHR

WB: poly/mono-ADPr/AHR

WB: DTX2

WB: α-tubulin

**C** A549 cells

DMSO · Tapinarof

siRNA: control, DTX2 #1, DTX2 #2, control, DTX2 #1, DTX2 #2

WB: poly/mono-ADPr

WB: AHR

WB: poly/mono-ADPr/AHR

WB: DTX2

WB: α-tubulin

**Figure 4. DTX2 mediates degradation of AHR and other ADP-ribosylated substrates upon AHR activation.**

(**A**) HCC44 cells were transfected with siRNAs against DELTEX E3 ligases and other ADP-ribose-binding E3 ligases, treated with DMSO or 1 μM tapinarof for 24 h, and analysed by western blotting. (**B**) HCC44 cells transfected with two separate DTX2 siRNAs, treated with DMSO or 1 μM tapinarof for 24 h, and analysed by western blotting. (**C**) A549 cells transfected with two separate DTX2 siRNAs, treated with DMSO or 1 μM tapinarof for 24 h, and analysed by western blotting. ADPr was assessed with a poly-mono-ADPr antibody, DTX2 knockdown was confirmed with an anti-DTX2 antibody, α-tubulin was used as a loading control. Overlap of the signals was obtained using fluorescently labelled anti-rabbit (poly/mono-ADPr, green) and anti-mouse (AHR, red) secondary antibodies. Source data are available online for this figure.

raising an intriguing possibility for the wider role of PARP7 in the ubiquitin pathway.

Taken together, we have shown that the extent of ADPr-targeted protein degradation is much larger than previously thought. Importantly, this has allowed us to answer the long-standing question about the degradation of transcriptionally active AHR and PARP7, both of which are important therapeutic targets in cancer, infections, and immune diseases (Gorelik et al, 2025; Polonio et al, 2025). Mass spectrometry-based proteomic approaches to map the ADPr sites that trigger protein degradation, will be instrumental in further determining the physiological roles of this phenomenon. Our results and tools we developed pave the way for the discovery of new signalling pathways controlled through the ADPr and ubiquitylation crosstalk.

## Methods

### Reagents and tools table

| Reagent/resource | Reference or source | Identifier or catalogue number |
|---|---|---|
| **Experimental models** | | |
| HCC44 (*Homo sapiens*) | DSMZ | ACC 534 |
| MCF7 (*Homo sapiens*) | ATCC | HTB-22 |
| MCF7 PARP7 KO (*Homo sapiens*) | Jason Matthews laboratory (Rasmussen et al, 2021) (1) | N/A |
| A549 (*Homo sapiens*) | ATCC | CCL-185 |
| A549 DTX2 KO (*Homo sapiens*) | This paper | N/A |
| CT26 (*Mus musculus*) | ATCC | CRL-2638 |
| CT26 HiBiT-PARP7 (*Mus musculus*) | Michael Cohen laboratory (Sanderson et al, 2023) (2) | N/A |
| **Antibodies** | | |
| Mouse anti-AHR | Santa Cruz | sc-133088 |
| Rabbit anti-alpha-tubulin | Proteintech | 11224-1-AP |
| Rabbit anti-poly/mono-ADP-ribose | Cell Signaling Technology | 89190 |
| Mouse anti-mono-ADP-ribose | Bio-Rad | AbD43647 |
| Rabbit anti-DTX2 | Novus biologicals | NBP1-53119 |
| Mouse anti-HiBiT | Promega | N7200 |
| Rabbit anti-PARP1 | Abcam | ab227244 |

| Reagent/resource | Reference or source | Identifier or catalogue number |
|---|---|---|
| Mouse anti-PARP7 | Jason Matthews laboratory (Rasmussen et al, 2021) (1) | N/A |
| Rabbit anti-PARP14 | Abcam | ab229756 |
| Mouse anti-TNKS | Santa Cruz | sc-365897 |
| Mouse anti-Ubiquitin | Santa Cruz | sc-8017 |
| Goat anti-rabbit IgG secondary (800CW) | Li-COR Biosciences | 926-32211 |
| Goat anti-mouse IgG secondary (680RD) | Li-COR Biosciences | 926-68070 |
| Goat anti-Rabbit IgG Alexa Fluor™ 488 | Thermo Fisher Scientific | A11034 |
| Donkey anti-Mouse IgG Alexa Fluor™ 647 | Thermo Fisher Scientific | A31571 |
| **Oligonucleotides and other sequence-based reagents** | | |
| DTX1 siRNA (*Homo sapiens*) | Life Technologies | s4356 |
| DTX2 #1 siRNA (*Homo sapiens*) | Life Technologies | s41519 |
| DTX2 #2 siRNA (*Homo sapiens*) | Life Technologies | s227390 |
| DTX3 siRNA (*Homo sapiens*) | Life Technologies | s47003 |
| DTX3L siRNA (*Homo sapiens*) | Life Technologies | s45595 |
| DTX4 siRNA (*Homo sapiens*) | Life Technologies | s23319 |
| RNF114 siRNA (*Homo sapiens*) | Life Technologies | s31752 |
| RNF125 siRNA (*Homo sapiens*) | Life Technologies | s29809 |
| RNF138 siRNA (*Homo sapiens*) | Life Technologies | s28153 |
| RNF146 siRNA (*Homo sapiens*) | Life Technologies | s37821 |
| RNF166 siRNA (*Homo sapiens*) | Life Technologies | s41901 |
| HUWE1 siRNA (*Homo sapiens*) | Life Technologies | s19595 |
| control siRNA | Life Technologies | 4390847 |
| qPCR primers | This study | Supplementary Table 1 |
| DTX2 gRNA sequence | This study | Materials and methods |

| Reagent/resource | Reference or source | Identifier or catalogue number |
|---|---|---|
| **Chemicals, enzymes and other reagents** | | |
| TAK243 | MedChemExpress | HY-100487 |
| RBN2397 | MedChemExpress | HY-136174 |
| Olaparib | LKT Laboratories | O4402 |
| RBN012759 | MedChemExpress | HY-136979 |
| AZ6102 | MedChemExpress | HY-12975 |
| MG132 | Sigma-Aldrich | 4747911 |
| tapinarof | MedChemExpress | HY-109044 |
| ISRIB | Cayman Chemical | 41879 |
| 4µ8c | APExBIO | B1874 |
| PDD00017273 | Sigma-Aldrich | SML1781 |
| **Software** | | |
| Odyssey Image Studio Lite® software v.5.2 | LI-COR Biosciences www.licorbio.com | |
| GraphPad Prism 10 | GraphPad www.graphpad.com | |
| Fiji | www.fiji.sc | |
| **Other** | | |
| Dynabeads™ Protein G | Invitrogen | 10003D |
| SensiMix SYBR® No-ROX | Bioline, Meridian Bioscience | QT650-05 |
| Hoechst 33342 | Thermo Fisher Scientific | H1399 |
| Lipofectamine™ RNAiMAX | Thermo Scientific | 13778150 |
| QuantiTect Reverse Transcription Kit | QIAGEN | 205311 |
| RNeasy Mini Plus Kit | QIAGEN | 74134 |
| Nano-Glo® HiBiT Lytic Detection System | Promega | N3030 |

## Cell culture and treatments

Human HCC44 cell line was purchased from the German Collection of Microorganisms and Cell Cultures (#ACC 534, DSMZ, Braunschweig, Germany). Human A549 (#CCL-185, ATCC) and MCF7 cells (#HTB-22, ATCC, wild type and PARP7 KO described previously (Rasmussen et al, 2021)) were cultured in DMEM medium (with GlutaMAX) supplemented with 10% FBS, and penicillin-streptomycin (100 U/ml and 100 µg/ml, respectively). HCC44 and mouse CT26 cells (#CRL-2638, ATCC, wild type and HiBiT-PARP7 have been described previously (Sanderson et al, 2023)) were cultured in RPMI1640 medium supplemented with 10% FBS, and penicillin-streptomycin (100 U/ml and 100 µg/ml, respectively). All cells were cultured in a humidified incubator at 37 °C with 5% (v/v) $CO_2$ atmosphere. The following drug concentrations were used: TAK243 (1 µM, HY-100487, MedChemExpress), RBN2397 (100 nM, HY-136174, MedChemExpress), olaparib (100 nM), RBN012759 (100 nM (or 500 nM in Fig. 2D), HY-136979, MedChemExpress), AZ6102 (1 µM, HY-12975, MedChemExpress), MG132 (10 µM, 4747911, Sigma-Aldrich),

tapinarof (1 µM, HY-109044, MedChemExpress), ISRIB (10 µM, 41879, Cayman Chemical), 4µ8c (10 µM, B1874, APExBIO). Cell lines were regularly tested for mycoplasma contamination.

## Western blotting

Cells were seeded in six-well plates (Corning). Once 80–90% confluent, cells were treated with compounds, washed in Dulbecco's phosphate-buffered saline, scraped and centrifuged at room temperature (3000 rpm for 4 min) to obtain pellets. Pellets were lysed immediately by resuspension in 50 µl lysis buffer (50 mM Tris pH 8, 100 mM NaCl, 1% Triton X-100 pH 8, protease and phosphatase inhibitor, 1 µM olaparib, 1 µM PARG inhibitor (PDD00017273, Sigma-Aldrich), 2 µM benzonase) and incubated on ice for 25 min to extract proteins. Protein concentration was determined using the Pierce™ BCA Protein Assay Kit (Thermo Fisher Scientific, #23225) according to the manufacturer's instructions and adjusted to 1 µg/µl by adding the required volume of MilliQ $H_2O$ and NuPAGE® LDS Sample Buffer with 25 mM Dithiothreitol (DTT) final concentration. Samples were then boiled at 95 °C for 5 min. Samples were loaded on a NuPAGE™ 4–12% Bis-Tris Gels (Invitrogen). Gels were subsequently subjected to SDS-PAGE at 160 V in MOPS buffer before transfer to a Trans-Blot Turbo Transfer membrane (Bio-Rad) using a Trans-Blot Turbo Transfer System (Bio-Rad) (standard 30 min transfer). Ponceau S staining (Sigma) was used to determine transfer efficiency. Membranes were blocked using 5% (w/v) bovine serum albumin (BSA) in 1X phosphate-buffered saline-0.1% (w/v) Tween® 20 (PBS-T) for 1 h. Membranes were subsequently incubated in primary antibodies overnight at 4 °C, followed by secondary antibody incubation for one hour at room temperature. Antibody list: AHR (Santa Cruz, sc-133088, 1:1000, mouse), alpha-tubulin (Proteintech, 11224-1-AP, 1:2000, rabbit), poly/mono-ADP-ribose (Cell Signaling Technology, #89190 (D9P7Z), 1:1000, rabbit), mono-ADP-ribose (HRP-conjugated, Bio-Rad, AbD43647), PARP7 (Jason Matthews laboratory, 1:1000, mouse), HiBiT (Promega, 1:1000, mouse, N7200), PARP1 (Abcam, 1:1000, rabbit, ab227244), TNKS (Santa Cruz, 1:500, mouse, sc-365897), PARP14 (Abcam, 1:1000, rabbit, ab229756), DTX2 (Novus biologicals, 1:1000, rabbit, NBP1-53119), goat anti-rabbit IgG secondary (800CW, 926-32211, Li-COR Biosciences), goat anti-mouse IgG secondary (680RD, 926-68070, Li-COR Biosciences). All antibodies were diluted in 5% (w/v) BSA in 1X PBS-T. Blots were visualised using the Odyssey CLx Imager (LI-COR Biosciences). Quantification was performed using the Odyssey Image Studio Lite® software v.5.2 (LI-COR Biosciences).

## HiBiT luminescence assay

HiBiT-PARP7 CT26 cells or wild-type CT26 cells (non-HiBiT-tagged control) were seeded in triplicates at 10,000 cells per well in 50 µl of medium in a 96-well clear bottom white plate. The following day, drug treatments were performed by adding 50 µl of 2x drug in culture medium. Following drug incubation, Nano-Glo® HiBiT Lytic Detection System was used (Promega, N3030) according to the manufacturer's instructions. The luminescence was quantified using the SpectraMax M5 plate reader (Molecular Devices).

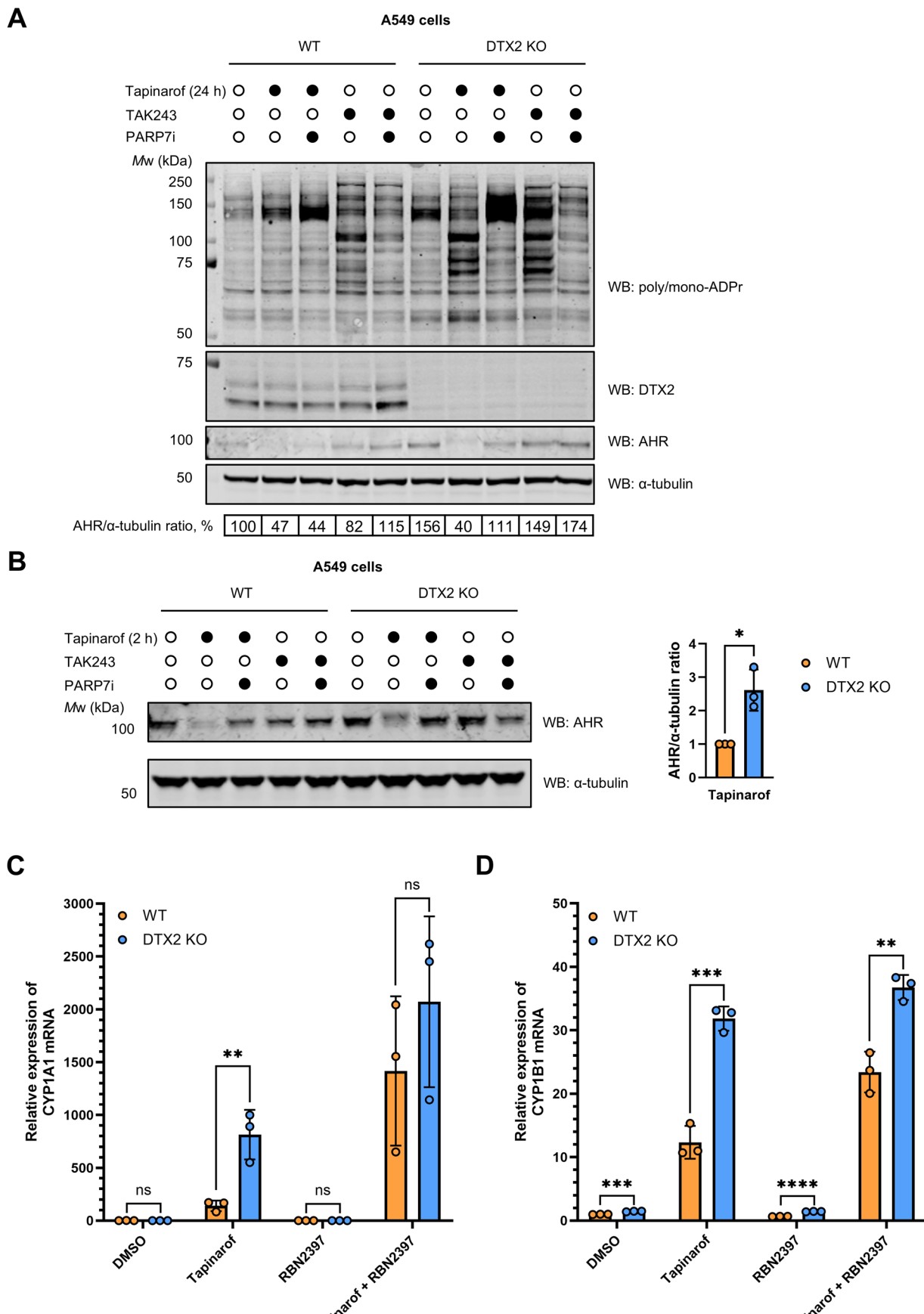

**Figure 5.  DTX2 knockouts display increased ADPr and AHR levels upon AHR activation.**

(**A**) Wild type and A549 DTX2 knockout (KO) cells were treated with 1 µM tapinarof (24 h), 100 nM RBN2397 (24 h) or 1 µM TAK243 (4 h) alone and in combinations. ADPr was assessed by western blotting using the anti-poly/mono-ADPr antibody. AHR and DTX2 were detected with anti-AHR and anti-DTX2 antibodies, respectively, with α-tubulin as a loading control. (**B**) A549 cells were treated as in (**A**), but 1 µM tapinarof treatment was for 2 h. AHR/α-tubulin ratio was quantified in the tapinarof condition. *$P$ = 0.0106, calculated by Student's $t$-test (two-tailed, unpaired). Data were shown as mean ± s.d. of $n$ = 3 biological replicates. (**C**) Wild type and A549 DTX2 knockout (KO) cells were treated with 1 µM tapinarof (24 h) or 100 nM RBN2397 (24 h) alone and in combination. DMSO was used as a vehicle control. CYP1A1 mRNA levels were measured using RT-qPCR. **$P$ = 0.0082, ns: no significant difference, calculated by Student's $t$-test (two-tailed, unpaired). Data were shown as mean ± s.d. of $n$ = 3 biological replicates. (**D**) CYP1B1 mRNA levels were measured using RT-qPCR, same conditions as in (**C**). **$P$ = 0.0036, ***$P$ = 0.0003 (DMSO), ***$P$ = 0.0005 (Tapinarof), ****$P$ < 0.0001, calculated by Student's $t$-test (two-tailed, unpaired). Data were shown as mean ± s.d. of $n$ = 3 biological replicates. Source data are available online for this figure.

## siRNA transfection

Cells were seeded at $8 \cdot 10^5$ cells per well in a six-well plate and reverse transfected with Lipofectamine™ RNAiMAX (Thermo Scientific) according to manufacturer's instructions. The following day, the medium was changed to fresh medium. The total transfection time was 48 h. The following Silencer™ Select siRNAs (Life Technologies) were used for transfection: DTX1 - s4356, DTX2 - s41519 (siRNA #1), s227390 (siRNA #2), DTX3 - s47003, DTX3L - s45595, DTX4 - s23319, RNF114 - s31752, RNF125 - s29809, RNF138 - s28153, RNF146 - s37821, RNF166 - s41901, HUWE1 - s19595, control siRNA No.2 - 4390847.

## Generation of DTX2 knockout A549 cells

A549 DTX2 knockout cells were generated by nucleofection of ribonucleoprotein (RNP) complexes as previously described (Groslambert et al, 2023), consisting of the Cas9 nuclease pre-loaded with sgRNAs:

DTX2_ex1 GGCGGTAGCGGCCTGACACGGGG
DTX2_in1 CCGGCTAAGAGCGGAAACACAGG.

Forty-eight hours after nucleofection, single cells were sorted into 96-well plates by FACS. Knockout clones were confirmed by western blotting.

## RT-qPCR

Total RNA was isolated using the RNeasy Mini Plus Kit (QIAGEN). cDNA was obtained with 500 ng RNA using a QuantiTect Reverse Transcription Kit (QIAGEN) according to the manufacturer's instructions. RT-qPCR was performed using the SensiMix SYBR® No-ROX (QT650-05, Bioline, Meridian Bioscience) on a Rotor-Gene Q instrument (QIAGEN). The relative gene expression analysis was performed using the ddCt method, normalised to hypoxanthine phosphoribosyltransferase 1 (*HPRT1*). Primers are listed in Appendix Table S1.

## Immunoprecipitation

Cells from 10 cm petri dishes were treated and lysed in lysis buffer (50 mM Tris, pH 8, 100 mM NaCl, 1% Triton X-100, pH 8, protease and phosphatase inhibitor, 1 µM olaparib, 1 µM PARG inhibitor (PDD00017273, Sigma-Aldrich), 2 µM benzonase) and incubated on ice for 30 min to extract proteins. The lysates were spun down for 10 min at 15,000 rpm at 4 °C. Supernatants were transferred into new tubes and incubated with an anti-AHR (3 µg), anti-HiBiT (3 µg) or a mouse IgG control (3 µg) antibody on a roller for 2 h at

4 °C. 15 µl of Dynabeads™ Protein G (10003D, Invitrogen™) were added to the lysate/antibody mixtures and incubated for a further 2 h at room temperature. The beads were washed three times with a wash buffer (10 mM Tris, pH 8.0, 150 mM NaCl, 0.5% Triton X-100) and bound antibody-protein complexes were eluted by boiling for 5 min in NuPAGE® LDS Sample Buffer with 25 mM Dithiothreitol (DTT) final concentration. Western blotting was performed as described above.

## Treatment of cell lysates with ADP-ribose hydrolases

HCC44 cell lysates were prepared as described above. About 3 µg of *S. pyogenes* macrodomain *Spy*MacroD (WT or C119T inactive mutant) or 3 µg ARH1 were added to 60 µg of lysate and incubated at 37 °C, 1000 rpm shaking for 2 h. Recombinant *Spy*MacroD and ARH1 were purified as described previously (Ariza et al, 2024; Rack et al, 2018).

## In vitro DTX2 reaction

Following immunoprecipitation, AHR bound to 15 µl of magnetic Dynabeads was incubated with 0.1 µM human UBA1, 1 µM UbcH5B, 10 µM ubiquitin, and 0.5 µM His-DTX2-RING-DTC (390-C) in 50 mM Tris HCl, pH 8.0, 50 mM NaCl, 2.5 mM MgCl$_2$, 2.5 mM ATP, 0.5 mM DTT, at 37 °C, 300 rpm shaking for 30 min. To stop the reaction, the supernatant was removed, and the beads were washed and resuspended in 20 mM HEPES, pH 7.5, 50 mM NaCl, 5 mM MgCl$_2$ and 0.5 mM DTT. For hydrolases treatment, the resuspended beads were split into equal aliquots and treated with buffer, 1 mM NH$_2$OH, 1 µM *Spy*MacroD or 1 µM USP2, at 37 °C, 300 rpm shaking for 30 min. The supernatant was collected, and the beads were eluted with 12.5 µl 2X LDS buffer with 150 mM DTT, incubated at room temperature for 10 min. Western blotting was performed as described above.

## Immunofluorescence

Cells were seeded in a glass-bottom 96-well Sensoplate (Greiner Bio-One), allowed to adhere overnight before being treated with tapinarof (1 µM) or PARP7i (RBN2397, 100 nM) for 24 h. TAK243 (1 µM) was added to cells for 4 h. Cells were stained as previously described (Kar et al, 2024). Briefly, growth medium was removed from wells and cells were washed once with PBS before being fixed with ice-cold methanol:acetone (1:1, v/v) and placed at −20 °C for 15 min. Fixative was then removed, and cells were washed three times with PBS, then blocked in blocking buffer (3% BSA, w/v, in PBS + 0.2% Tween® 20). Cells were incubated in primary

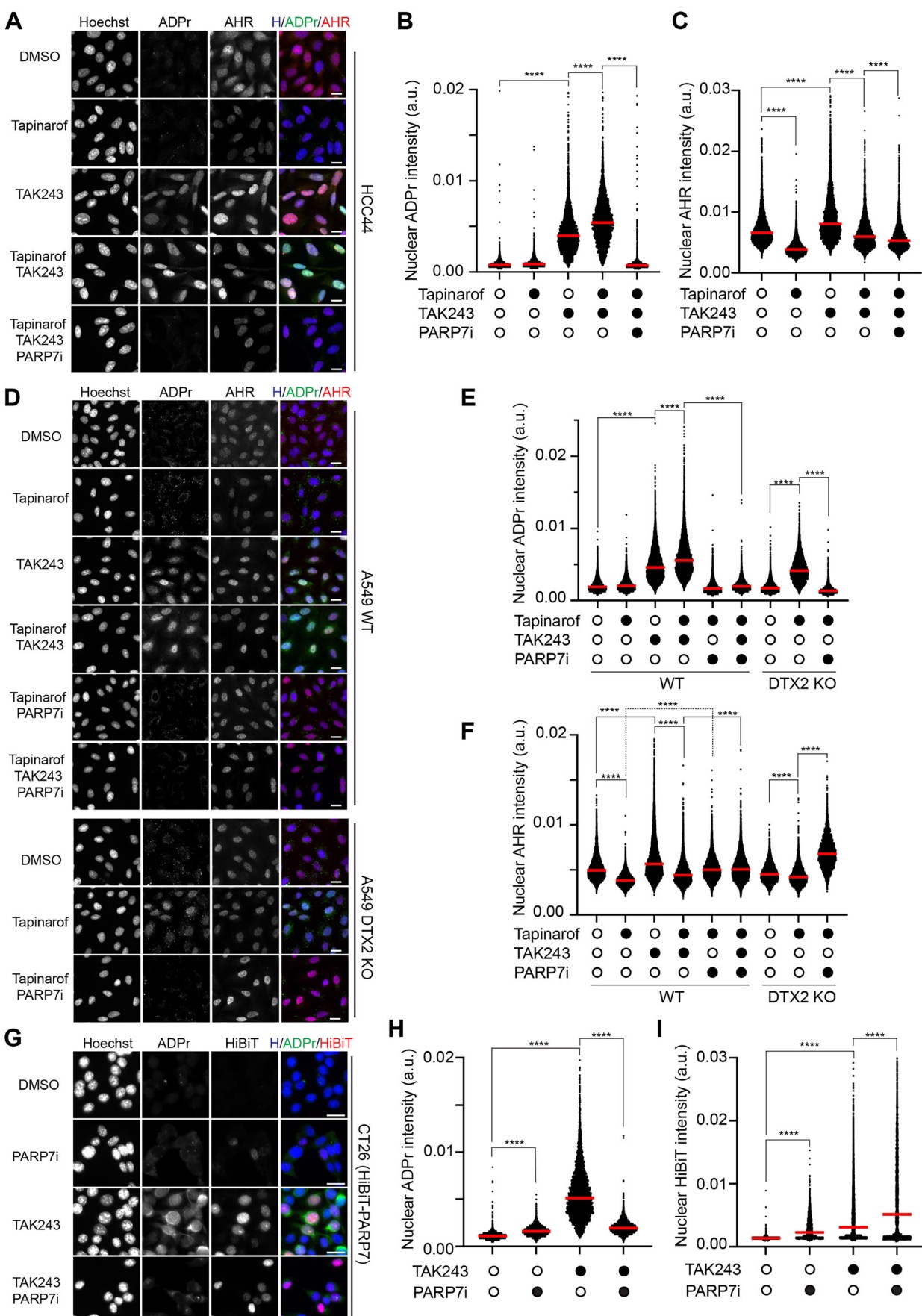

**Figure 6. ADP-ribose, AHR and PARP7 are enriched in the nucleus upon TAK243/tapinarof treatments.**

(A) Widefield images showing HCC44 cells treated with 1 µM tapinarof (24 h), 1 µM TAK243 (4 h) and 100 nM PARP7i (RBN2397, 24 h) alone and in combination as in Fig. 3B. Cells were stained with Hoechst (blue), ADPr (Poly/mono-ADP-ribose, green) and AHR (magenta). (B, C) Quantification of nuclear ADPr (B) or AHR (C) from (A). Between 5156 and 6220 cells were measured in each condition. (D) Widefield images showing A549 WT or DTX2 knockout (KO) cells treated with 1 µM tapinarof (24 h), 1 µM TAK243 (4 h) and 100 nM PARP7i (RBN2397, 24 h) alone and in combination as in Fig. 5A. Cells were stained with Hoechst (blue), ADPr (poly/mono-ADP-ribose, green) and AHR (magenta). (E, F) Quantification of nuclear ADPr (E) or AHR (F) from (D). Between 5716 and 13282 cells were measured in each condition. (G) Widefield images showing CT26 (HiBiT-PARP7) cells treated with 1 µM TAK243 (4 h) and 100 nM PARP7i (RBN2397, 24 h) alone and in combination, as in Fig. 2B. Cells were stained with Hoechst (blue), ADPr (poly/mono-ADP-ribose, green) and HiBiT (magenta). (H, I) Quantification of nuclear ADPr (B) or HiBiT (I) from (G). Between 2991 and 6701 cells were measured in each condition. For all images, scale bar = 20 µm. Statistical analysis was performed using an ordinary one-way ANOVA. Asterisks indicate statistical significance (****$P$ < 0.0001). Red bars indicate the median for each condition. Source data are available online for this figure.

antibodies diluted in blocking buffer overnight at 4 °C (poly/mono-ADP-ribose (D9P7Z), Cell Signaling technology, 1:500; Ubiquitin, SC-8017, Santa Cruz, 1:100; Aryl Hydrocarbon Receptor, SC-133088, Santa Cruz, 1:100; HiBiT, N7200, Promega, 1:1000). Primary antibody was removed from wells, and the cells were washed three times with PBS + 0.1% Triton X-100. Secondary antibody (Goat anti-Rabbit IgG Alexa Fluor™ 488 A11034 and Donkey anti-Mouse IgG Alexa Fluor™ 647 A31571) was diluted 1:500 in blocking buffer with 1.25 µg/mL Hoechst 33342 (Thermo Fisher Scientific) for 1 h at room temperature with gentle rocking. Cells were washed three times with PBS + 0.1% Triton X-100 prior to imaging. Cells were imaged on an EVOS M7000 widefield microscope equipped with a UPlanSApo 20x/0.75 N.A. objective lens, DAPI, GFP, and Cy5 EVOS™ Light Cubes and a CMOS camera. Images were analysed using CellProfiler (Stirling et al, 2021) to measure mean intensity grey values in the nucleus of cells. Experiments were repeated a minimum of two times.

## Statistical analysis

Statistical analysis was performed in GraphPad Prism 10. An unpaired two-tailed Student's $t$-test was used to analyse differences between two groups (RT-qPCR and western blot quantification). An ordinary one-way ANOVA was used to analyse differences between multiple groups (microscopy). $P$ values of less than 0.05 were considered significant. All experiments were reproduced at least twice, with a representative image of the replicates shown for western blots. No statistical method was used to predetermine the sample size.

## Data availability

All data needed to support the conclusions of the paper are included within the main text and in the appendix. Source immunofluorescence and western blotting data have been uploaded to the BioImage Archive with the accession number S-BIAD2306.

The source data of this paper are collected in the following database record: biostudies:S-SCDT-10_1038-S44318-025-00656-1.

## Peer review information

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

## Acknowledgements

A.G. was funded by the Sir Henry Wellcome Fellowship (Wellcome Trust, 224095/Z/21/Z). The work in I.A.'s laboratory is supported by the Wellcome Trust (210634, 223107 and 302632); Biotechnology and Biological Sciences Research Council (BB/R007195/1 and BB/W016613/1) and Cancer Research United Kingdom (C35050/A22284). We would like to thank Dr Alan Wainman and the Dunn School Bioimaging Facility for expert advice and access to the microscope. We thank Vasiliki Tsioligka at the Don Mason Facility of Flow Cytometry for help with single-cell sorting. We would also like to thank Dr Joey Riepsaame and the Genome Engineering Oxford (GEO) Facility for help with DTX2 knockout cells. HiBiT-PARP7 CT26 cells were a kind gift from Prof. Michael S. Cohen (Oregon Health and Science University, USA).

## Author contributions

**Andrii Gorelik**: Conceptualisation; Data curation; Formal analysis; Supervision; Funding acquisition; Validation; Investigation; Visualisation; Methodology; Writing—original draft; Project administration; Writing—review and editing. **Nina Đukić**: Validation; Investigation; Visualisation; Methodology. **Rebecca Smith**: Data curation; Formal analysis; Investigation; Visualisation; Methodology. **Chatrin Chatrin**: Investigation; Methodology. **Osamu Suyari**: Investigation; Methodology. **Jason Matthews**: Resources; Methodology. **Ivan Ahel**: Conceptualisation; Supervision; Funding acquisition; Writing—original draft; Project administration; Writing—review and editing.

Source data underlying figure panels in this paper may have individual authorship assigned. Where available, figure panel/source data authorship is listed in the following database record: biostudies:S-SCDT-10_1038-S44318-025-00656-1.

## Disclosure and competing interests statement

JM is a consultant for Duke Street Bio Inc. and NOA Therapeutics Inc. The remaining authors declare no competing interests.

