## [Peer Review File · The EMBO Journal]

Ubiquitin pathway blockade reveals endogenous ADP-ribosylation marking PARP7 and AHR for degradation

Andrii Gorelik, Nina Đukic, Rebecca Smith, Chatrin Chatrin, Osamu Suyari, Jason Matthews, and Ivan Ahel

Corresponding author(s): Ivan Ahel (ivan.ahel@path.ox.ac.uk) , Andrii Gorelik (andrii.gorelik@path.ox.ac.uk)

Review Timeline:

Submission Date:	22nd Jun 25
Editorial Decision:	21st Jul 25
Revision Received:	24th Sep 25
Editorial Decision:	27th Oct 25
Revision Received:	16th Nov 25
Accepted:	18th Nov 25

Editor: Hartmut Vodermaier

Transaction Report:

Dr. Ivan Ahel
University of Oxford
Sir William Dunn School of Pathology
South Parks Road
Oxford OX1 3RE
United Kingdom

21st Jul 2025

Re: EMBOJ-2025-121671
Ubiquitin pathway blockade reveals endogenous ADP-ribosylation marking PARP7 and AHR for degradation

Dear Ivan,

Thank you again for submitting your manuscript on endogenous AHR and PARP7 ADP-ribosylation for our consideration. Three expert referees have now assessed it, and provided the comments copied below. Since all reviewers consider the work timely and potentially interesting, we would be happy to pursue a revised version further for EMBO Journal publication. Nevertheless, the reports bring up a number of important concerns, which would need to be adequately addressed prior to acceptance. In particular, the referees ask for more decisive (bio)chemical characterization of the resulting post-translational modification(s), and for at least some follow-up on their functional consequences; as well as for various specific controls to validate certain approaches and investigate alternative scenarios.

Since it is our policy to allow only a single round of major revision, I would very much encourage you to contact me with a revision plan and preliminary point-by-point response already during the early stages of your revision work, so that we could discuss if and how the main points could be resolved; or whether a less completely revised manuscript might alternatively become suitable for publication in one of our sister journals like EMBO Reports or Life Science Alliance. We would also be open to extending the revision deadline if that should be helpful. Our scooping protection policy means that competing manuscripts published while your work is under revision will not have a negative effect on our final decision.

Detailed information on preparing, formatting and uploading a revised manuscript can be found below and in our Guide to Authors. Thank you again for the opportunity to consider this work for The EMBO Journal, and I look forward to hearing from you in due time.

With kind regards,

Hartmut

3) Revised manuscript text (including main tables, and figure legends for main and EV figures) has to be submitted as editable

text file (e.g., .docx format). We encourage highlighting of changes (e.g., via text color) for the referees' reference.

4) Each main and each Expanded View (EV) figure should be uploaded as individual production-quality files (preferably in .eps, .tif, .jpg formats). For suggestions on figure preparation/layout, please refer to our Figure Preparation Guidelines:

8) Please note that supplementary information at EMBO Press has been superseded by the 'Expanded View' for inclusion of additional figures, tables, movies or datasets; with up to five EV Figures being typeset and directly accessible in the HTML version of the article. For details and guidance, please refer to:

embopress.org/page/journal/14602075/authorguide#expandedview

9) To facilitate reproducibility and cross-laboratory adoption of methodologies, please structure the Materials & Methods section as outlined in our guide to authors, including a completed Reagents and Tools Table that can be downloaded from our author guidelines as well (<https://www.embopress.org/page/journal/14602075/authorguide#structuredmethods>).

10) Digital image enhancement is acceptable practice, as long as it accurately represents the original data and conforms to community standards. If a figure has been subjected to significant electronic manipulation, this must be clearly noted in the figure legend and/or the 'Materials and Methods' section. The editors reserve the right to request original versions of figures and the original images that were used to assemble the figure. Finally, we generally encourage uploading of numerical as well as gel/blot image source data; for details see: embopress.org/page/journal/14602075/authorguide#sourcedata

Further information is available in our Guide For Authors:

In the interest of ensuring the conceptual advance provided by the work, we recommend submitting a revision within 3 months (19th Oct 2025). Please discuss the revision progress ahead of this time with the editor if you require more time to complete the revisions. Use the link below to submit your revision:

Link Not Available

Referee #1:

EMBOJ-2025-121671 by Gorelik et al. (AheI)

"Ubiquitin pathway blockade reveals endogenous ADP-ribosylation marking PARP7 and AHR for degradation"

Summary

In this manuscript, the authors use two inhibitors of the ubiquitin-proteasome system, TAK243 and MG132, to uncover endogenous ADP-ribosylation by Western blotting. TAK243- and MG132-dependent ADP-ribosylation is mediated by PARP1, PARP7, and TNKS1/2 in A549 cells, and primarily mediated by PARP7 in HCC46, CT29, and MCF7 cells. The authors use the AHR agonist Tapanirof to demonstrate that AHR activation leads to AHR degradation through the combined ADP-ribosylation and ubiquitylation activities of PARP7 and DTX2. Imaging studies reveal that these actions occur in the nucleus. Overall, this manuscript provides evidence for a post-translational mechanism by which PARP7 and DTX2 regulated AHR stability following AHR activation.

Review

This manuscript is timely and interesting, and will be of interest to the field. It builds upon previous work indicating PARP7

regulates AHR stability following AHR activation. The use of TAK243 to increase levels of endogenous ADP-ribosylation is interesting and novel. Given that AHR and PARP7 are targets for cancer therapy, the findings here may be useful for future translational studies. Two weaknesses described below should be addressed before publication.

Strengths: The main conclusions of the work are supported by several experiments across multiple cell lines. The manuscript is well written and the data are presented clearly. The discussion section does a good job of placing these observations in the context of previous literature (though reference to the recently discovered MAR-ubiquitin hybrid PTM is notably absent) and provides interesting speculations about potential future directions.

Weaknesses: Given the recent emergence of the MAR-ubiquitin hybrid modification, additional characterization of the post-translational modification(s) mediated by PARP7 and DTX2 on AHR are warranted. The data are entirely immunoassays-evaluation of a functional outcome, such as AHR target gene expression, would strengthen the conclusions of the manuscript.

Comments:

Major comments

(1) This study presents the intriguing possibility that endogenous MARYlation may be more widespread and abundant than previously appreciated (as demonstrated by the apparent quantitative MARYlation of AHR in Tapinarof + TAK243 treated cells) because some MARYlated proteins are rapidly degraded and therefore may have been missed in previous studies of MARYlation that did not use inhibitors of the ubiquitin-proteasome system.

Recent studies have found that PARPs and Deltex proteins (including PARP7 and DTX2) combine their enzymatic activities to create protein-conjugated MAR-ubiquitin modifications through non-canonical O-linked ubiquitination of the adenosine ribose within MAR.

As it stands, the data indicate that MARYlated AHR is rapidly modified with ubiquitin, but the nature of the AHR ubiquitin modification(s) remain unclear. Possibilities include: 1) ubiquitylation of AHR is the non-canonical O-linked form attached to MAR 2) ADPr promotes the interaction between DTX2 and AHR, leading to canonical Ub on AHR lysine residues or 3) a combination of the two.

Additional characterization of the AHR ubiquitin modification(s) is warranted. For example, does the poly/mono-ADPr detection reagent used in this study recognize MAR and MAR-ubiquitin with equal affinity? If not, this may explain the differences in signal with this reagent on blots of TAK243 vs. MG132 treated lysates. Furthermore, a variety of chemical/enzymatic treatments can be used to distinguish between canonical and ADPr-linked Ub. The authors should use these methods to qualitatively measure which types of Ub modifications are present on AHR.

(2) The authors demonstrate that the stability of HiBiT-PARP7 is regulated by PARP7 activity and the ubiquitin-proteasome system in Figure 2B and 2C. However, in Figure 2A, 3B, and 3C, only PARP7 inhibition with RBN2397 treatment stabilizes endogenous PARP7. Is the discrepancy between HiBiT-PARP7 and endogenous PARP7 because HiBiT-PARP7 is overexpressed in the CT26 cell line? Does this suggest there is a ubiquitin-independent mechanism regulating the stability of endogenous PARP7?

(3) DTX2 presumably recognizes the mono-ADP-ribosylation formed on AHR by PARP7 and thus regulating AHR stability. Can the authors clarify whether the interaction between DTX2 and AHR is dependent on MARYlation / PARP7 catalytic activity? As the authors point out, DTX2 knockdown stabilized AHR in the absence of Tapinarof (Figure 4C) when AHR is presumably in the cytoplasm and not engaged with nuclear PARP7.

(4) The molecular findings in this manuscript could be of broad interest to cancer biologists. For example, recent studies have demonstrated synergy between Tapinarof and RBN2397 across a several cancer cell lines (reference 11 and 14 in the manuscript). Tapinarof + RBN2397 also leads to higher transcription of AHR target genes (e.g., CYP1A1) than Tapinarof alone. Evaluating the effect of the combination of treatments used here (Tapinarof, RBN2397, TAK243, MG132) on AHR target gene expression would provide some insight into how the protein modifications described here might affect gene expression and cellular outcomes in cancers.

Minor comments

(1) In the Western blot, is the ~100 kDa mono-ADP-ribosylation signal detected corresponding to AHR mono-ADP-ribosylation? If so, could the authors explain why there is no increase in AHR ADP-ribosylation upon DTX2 knockdown under DMSO-treated conditions in Figures 4B and 4C?

(2) In Figure 5, even when DTX2 is knocked out, treatment with tapinarof appears to induce AHR degradation (lane 7). Does this suggest that E3 ligases other than DTX2 are involved in regulating AHR stability?

Referee #2:

Detecting endogenous ADP-ribosylation caused by specific PARP family members is difficult. The manuscript by Gorelik et al. shows that blocking the ubiquitin pathway (using the E1 inhibitor TAK-243 or the proteasome inhibitor MG132) enables the detection of endogenous ADP-ribosylation in various cell types. The main bands detected are approximately 75 kDa and 100

kDa, which match the molecular weight of PARP7 and the aryl hydrocarbon receptor (AHR), a transcription factor previously shown to be ADP-ribosylated by PARP7. Systematic testing with different selective PARP inhibitors demonstrates that PARP7 activity is necessary for the ADP-ribosylation of these bands. In HCC44 and MCF7 cells, the AHR agonist tapinarof increases ADP-ribosylation of these bands, and pharmacological and knockout (KO) studies further confirm that this modification depends on PARP7. A siRNA-mediated knockdown (KD) screen of various E3 ligases revealed that KD of DTX2 enhanced the ADP-ribosylation of these two bands. Reducing or eliminating DTX2 raised AHR levels. This, along with the increased ADP-ribosylation of AHR, led the authors to propose that PARP7-mediated ADP-ribosylation of AHR (and possibly other PARP7 targets) promotes its degradation.

The manuscript is timely, reflecting the rising interest in the interaction between ADP-ribosylation and ubiquitination. However, there are a few issues that need clarification to support the study's claims.

Major comments:

1. The authors interpret the TAK243-dependent detection of endogenous ADP-ribosylation as being due to the stabilization of the ADP-ribosylated protein. Since TAK243 (and MG132) induce a stress response (both can induce an ER stress response), it is possible that the reason why ADP-ribosylation is detected is because it is caused by activation of ER stress. The authors should determine if blocking ER stress prevents ADP-ribosylation.
2. A major claim in the paper is that the ADP-ribosylated bands at 75 kDa and 100 kDa are PARP7 and AHR, respectively. The authors should substantiate this claim, for example, by doing IP experiments. Alternatively, the authors can perform immunodepletion experiments in lysates.
3. In Fig. 3, the authors state that the "shift" in AHR after AHR agonist treatment results from ADP-ribosylation. While using a PARP7 inhibitor prevents this shift, supporting their claim, it is also possible that the shifted band indicates phosphorylated AHR. In fact, AHR agonists are known to cause AHR phosphorylation and affect its nuclear import (PMID: 37696456). If the authors claim that the shifted band is ADP-ribosylated AHR, they should provide supporting evidence for this assertion. Otherwise, they should temper their claim and consider other PTMs like phosphorylation.
4. In Fig. 4A, the controls showing that the siRNAs are functioning are missing. Additionally, the authors should use PARP7i and TNK1/2i to demonstrate that the DTX2 KD-induced ADP-ribosylation of the 75 kDa band is mediated by PARP7 and that the two bands around 150 kDa depend on TNK1/2.
5. In Fig 4B, C, DTX2 KD in the presence of the AHR agonist does not appear to rescue AHR levels. Do the authors have an explanation for this? In the presence of the AHR agonist, the authors state that PARP7 quantitatively ADP-ribosylates AHR (Fig. 3). If PARP7-mediated degradation of AHR is dependent on DTX2, should DTX2 knockdown prevent the AHR agonist-mediated decrease in AHR levels? The authors should address this experimentally or in the text.
6. In Fig. 5, it is unclear whether PARP7i prevents AHR degradation. This could suggest that an AHR agonist induces AHR degradation in a manner that is independent of PARP7. The authors should quantify these results for clarity. Additionally, in the context of DTX2 KO, AHR levels stay lower even when an AHR agonist is present. If DTX2 regulates AHR stability, I expect AHR levels to be higher in this condition. Can the authors explain this?
7. In Fig. 6F, the authors should determine if PARP7i impacts nuclear AHR levels upon AHR agonist treatment in WT cells.
8. In Fig. 6, why do nuclear AHR levels in the presence of AHR agonist decrease even in the presence of TAK243? Is there a change in localization (it's hard to discern from the images).
9. In Fig. S1, it appears that PARP7i partially restores the TAK243-dependent decrease in total nuclear ubiquitin. This is an interesting result that the authors should further discuss in the text. In any event, the authors should provide a statistical analysis.

Minor comments:

1. The authors should specify in the figure legend the specific inhibitors used and their concentrations, as these are included in the methods but are better placed in the legend for easier access by the reader.
2. Page 15, Line 243: "bot" should be "blot"

Referee #3:

In this study, Gorelik and colleagues investigate cellular ADP-ribosylation primarily in the context of PARP7 and AHR (aryl hydrocarbon receptor). An innovative aspect of the study is the use of the inhibitor TAK243 to inactivate the ubiquitylation system and thereby elevate a mono-ADP-ribose signal that is normally transient. The most prominent increase in mono-ADP-ribose signal was focused in a few bands in Western blots, and the study focused on the 100 kDa and 75 kDa bands. The presented experiments together with existing literature gives strong evidence that these major bands are AHR and PARP7, and they also use the compound tapinarof as a stimulator of AHR signalling. Through a knockdown screen of E3 ligases, the work identifies DTX2 as the major ubiquitin ligase involved in AHR and PARP7 turnover in this system. Imaging of cells under different treatment conditions matches well to the observations made by Western blot analysis. The experiments are nicely presented and well controlled, and the conclusions match the results and are consistent with previous literature that is cited in the study, including work from the same group. The results are timely since other reports are appearing that connect mono-ADP-ribose modifications with ubiquitylation and specific E3 ligases, and I believe these are appropriately cited in the current work. I have a few minor points that might help improve the presentation of the study.

- TAK243 was used at 1 μ M according to the methods. Were other concentrations attempted and this found to be optimal? This study will likely serve as a resource for the clever trick of inhibiting ubiquitylation, so it could be helpful to report whether any optimization of conditions was necessary.

-page 6, lines 99,100. Suggest to find a different term than colocalise for the experiments trying to determine whether the 75 kDa band is indeed PARP7. Overlapping migration?

-Prior to the multi-color experiments in Figure 4, the shift in AHR migration is not always noticeable in the gel images. Is there a way to highlight the difference with gel markings?

-The discussion and some of the references refer to an ADP-ribose-ubiquitin dual modification. Is there any evidence that this is the modification that arises from PARP7 and DTX2? It would be useful to have at least some discussion on this point.

We thank all reviewers for taking the time to read our manuscript and for their constructive comments and useful suggestions that helped us improve it.

Referee #1:

1. Recent studies have found that PARPs and Deltex proteins (including PARP7 and DTX2) combine their enzymatic activities to create protein-conjugated MAR-ubiquitin modifications through non-canonical O-linked ubiquitination of the adenosine ribose within MAR. As it stands, the data indicate that MARYlated AHR is rapidly modified with ubiquitin, but the nature of the AHR ubiquitin modification(s) remain unclear. Possibilities include: 1) ubiquitylation of AHR is the non-canonical O-linked form attached to MAR 2) ADPr promotes the interaction between DTX2 and AHR, leading to canonical Ub on AHR lysine residues or 3) a combination of the two. Additional characterization of the AHR ubiquitin modification(s) is warranted. For example, does the poly/mono-ADPr detection reagent used in this study recognize MAR and MAR-ubiquitin with equal affinity? If not, this may explain the differences in signal with this reagent on blots of TAK243 vs. MG132 treated lysates. Furthermore, a variety of chemical/enzymatic treatments can be used to distinguish between canonical and ADPr-linked Ub. The authors should use these methods to qualitatively measure which types of Ub modifications are present on AHR.

We thank the reviewer for highlighting the exciting new area of ADPr-ubiquitin hybrid modification. While we recognise the potential of this modification occurring on AHR, the ADPr-Ub field is still in the early days in terms of detection methods and the lability of the modification prevents its reliable detection, especially given the multiple cellular hydrolases acting against this modification. To circumvent this, we immunoprecipitated AHR from HCC44 cell lysates and performed *in vitro* DTX2 assays (new Supplementary Fig. 7A). The DTX2 reaction resulted in a band shift of ADPr AHR. Hydroxylamine, treatment collapsed the bands, as did the USP2 treatment. SpyMacroD removed the ADPr signal consistent with its Glu/Asp/Cys activity.

It is possible that ubiquitin could prevent ADPr detection with commercially available ADPr antibodies. We have also added the data showing that pan-specific deubiquitinase USP2 treatment does not increase ADPr signal in MG132 treated samples (Supplementary Fig. 7B), suggesting that it is difficult to detect DTX2-dependent transient ubiquitination of ADPr in cell extracts.

While ADPr sites on AHR have not been validated we have additionally used a PARP7 Cys39 ADPr peptide to further characterise the ADPr-Ub modification. We

subjected the peptide to the reaction with recombinant DTX2 which resulted in a strong Ub shift and then performed a variety of treatments to test its stability. The poly/mono-ADPr antibody (D9P7Z, Rabbit mAb #89190, Cell Signalling Technology) appears to recognise the ADPr within the ADPr-Ub hybrid, but it also non-specifically recognises several other proteins in the mixture, for example recombinant hydrolases in the reaction, E1, E2 and an unidentified ubiquitin adduct. This suggests that while the antibody probably detects the correct signal in principle, improved specific antibodies need to be developed in the future for the reliable detection.

DTX2 reaction on a synthetic PARP7 peptide and its treatment with various hydrolases. 50 μ M PARP7 peptide (H-ITPLKTCFK-OH) with or without ADP-

ribosylation at C39 was incubated with 0.5 μM human UBA1, 2.5 μM UbcH5B, 10 μM ubiquitin, and 2 μM His-DTX2-RING-DTC (390-C) in 50 mM Tris HCl pH 8.0, 50 mM NaCl, 2.5 mM MgCl_2 , 2.5 mM ATP, 0.5 mM DTT, at 37°C for 30 minutes. For hydrolases treatment, 1 μM of the indicated enzymes were added, except for NudT16 (10 μM), Cezanne (20 nM), and hydroxylamine (1M). Reactions were carried out at 37°C for 30 minutes, and stopped by adding 1x LDS + 150 mM DTT. For hydrolases treatment, 1 μM of the indicated enzymes were added, except for NudT16 (10 μM), Cezanne (20 nM), and hydroxylamine (1M). Reactions were analysed by western blotting using poly/mono-ADPr (D9P7Z, Rabbit mAb #89190, Cell Signalling Technology) and ubiquitin (P4D1, sc-8017, Santa Cruz) antibodies.

2. The authors demonstrate that the stability of HiBiT-PARP7 is regulated by PARP7 activity and the ubiquitin-proteasome system in Figure 2B and 2C. However, in Figure 2A, 3B, and 3C, only PARP7 inhibition with RBN2397 treatment stabilizes endogenous PARP7. Is the discrepancy between HiBiT-PARP7 and endogenous PARP7 because HiBiT-PARP7 is overexpressed in the CT26 cell line? Does this suggest there is a ubiquitin-independent mechanism regulating the stability of endogenous PARP7?

We have performed qPCR in HCC44 cells treated with DMSO or PARP7i and detected a substantial increase in PARP7 mRNA. This suggests that PARP7 inhibition can trigger a transcriptional response that upregulates its mRNA levels, resulting in PARP7 detection only in conditions where PARP7i is used (in HCC44 cells). Mouse cells such as CT26 may have a different mechanism for PARP7 regulation. Furthermore, despite not being overexpressed, the HiBiT tag on the N-terminus of PARP7 may affect its stability.

RT-qPCR analysis of measure PARP7 mRNA levels upon PARP7i in HCC44 cells. Unpaired t test was conducted to show statistical significance; **** $P < 0.0001$. Primers used: Forward: TCTCAGGAGCACTTGGAAAGA; Reverse: GTGTGGACAGCCTTCGTAGT . mRNA expression was normalised to the *HPRT1* control gene.

3. DTX2 presumably recognizes the mono-ADP-ribosylation formed on AHR by PARP7 and thus regulating AHR stability. Can the authors clarify whether the interaction between DTX2 and AHR is dependent on MARylation / PARP7 catalytic activity? As the authors point out, DTX2 knockdown stabilized AHR in the absence of Tapanirof (Figure 4C) when AHR is presumably in the cytoplasm and not engaged with nuclear PARP7.

To address this, we have performed pulldowns of AHR and blotted for DTX2. Supplementary Fig. 8 shows that despite being weak, the interaction between AHR and DTX2 is ADPr-dependent. TAK243 induces the interaction while PARP7i treatment abolishes it. Regarding the localisation, AHR is primarily localised in the nucleus as shown in Fig. 6. Endogenous ligands such as kynurenine can promote its shuttling from the cytoplasm into the nucleus (Fig. 4A in Liu, Y., Liang, X., Yin, X. et al. Blockade of IDO-kynurenine-AhR metabolic circuitry abrogates IFN- γ -induced immunologic dormancy of tumor-repopulating cells. *Nat Commun* 8, 15207 (2017) <https://doi.org/10.1038/ncomms15207>)

4. The molecular findings in this manuscript could be of broad interest to cancer biologists. For example, recent studies have demonstrated synergy between Tapinarof and RBN2397 across a several cancer cell lines (reference 11 and 14 in the manuscript). Tapanirof + RBN2397 also leads to higher transcription of AHR target genes (e.g., CYP1A1) than Tapanirof alone. Evaluating the effect of the combination of treatments used here (Tapanirof, RBN2397, TAK243, MG132) on AHR target gene expression would provide some insight into how the protein modifications described here might affect gene expression and cellular outcomes in cancers.

We thank the reviewer for the useful suggestion to add functional significance to the DTX2-mediated AHR turnover. Due to proteome-wide effects, TAK243 and MG132 could affect many different pathways in the cell (including enzymes and transcription factors), complicating the interpretation of the transcription data. To prevent this, we have performed RT-qPCR in wild type and DTX2 knockout A549 cells treated with Tapinarof and PARP7i (alone or in combination) as a more targeted system. We measured CYP1A1 and CYP1B1 mRNA expression which are two classic AHR target genes (Fig. 5C and D). Interestingly, DTX2 knockouts exhibited higher CYP1A1 and CYP1B1 mRNA levels (six- and three-fold respectively) when treated with tapinarof compared to wild type cells. This fits with the mechanism for the rapid shutdown of AHR transcription via PARP7 and DTX2.

5. In the Western blot, is the ~100 kDa mono-ADP-ribosylation signal detected corresponding to AHR mono-ADP-ribosylation? If so, could the authors explain why there is no increase in AHR ADP-ribosylation upon DTX2 knockdown under DMSO-treated conditions in Figures 4B and 4C?

We have confirmed that the 100 kDa ADP-ribosylation signal belongs to AHR by performing immunoprecipitation (Supplementary Fig. 3).

Highest AHR ADP-ribosylation appears to occur when AHR is activated with a ligand. Endogenous tryptophan metabolites may activate AHR. In the TAK243-treated condition metabolic enzymes can be upregulated to produce these AHR ligands, resulting in AHR ADP-ribosylation. Moreover, under the DMSO treated conditions, the amount and/or activity of PARP7 may be insufficient to ADP-ribosylate AHR. In addition, hydrolases that remove ADP-ribose on AHR may have differential activity (suppressed) in the tapinarof condition. Thus, in DTX2 knockdowns in a steady state (DMSO treatment) there is no substantial increase in ADP-ribosylation on AHR compared to wild type cells.

6. In Figure 5, even when DTX2 is knocked out, treatment with tapinarof appears to induce AHR degradation (lane 7). Does this suggest that E3 ligases other than DTX2 are involved in regulating AHR stability?

We performed tapinarof treatments for 24 h and reasoned that during this time, other E3 ligases may indeed regulate AHR stability thereby reducing its levels in DTX2 KO cells to the same level as in wild type cells. To address this, we performed a shorter treatment of 2 h. With the shorter treatment AHR levels in DTX2 knockouts are substantially higher suggesting that DTX2-mediated degradation of AHR happens quickly. The new data are presented in Fig. 5B. This also addresses a similar concern from Referee #2.

Referee #2:

1. The authors interpret the TAK243-dependent detection of endogenous ADP-ribosylation as being due to the stabilization of the ADP-ribosylated protein. Since TAK243 (and MG132) induce a stress response (both can induce an ER stress response), it is possible that the reason why ADP-ribosylation is detected is because it is caused by activation of ER stress. The authors should determine if blocking ER stress prevents ADP-ribosylation.

We thank the reviewer for highlighting the possible ER stress-induced ADP-ribosylation. We used ER stress inhibitors to test this. Treatment of cells with distinct inhibitors of the ER stress pathway did not substantially affect ADPr levels of the 75 and 100 kDa bands (Supplementary Fig. 2). Furthermore, our findings suggest that in DTX2-depleted cells (that should not exhibit ER stress) treated with tapinarof, the ADPr pattern is very similar to TAK243-treated cells.

2. A major claim in the paper is that the ADP-ribosylated bands at 75 kDa and 100 kDa are PARP7 and AHR, respectively. The authors should substantiate this claim, for example, by doing IP experiments. Alternatively, the authors can perform immunodepletion experiments in lysates.

We thank the reviewer for suggesting this important experiment. We have now performed IP experiments to pull down AHR in HCC44 cells and blotted for ADPr. We have also performed PARP7 IP in HiBiT CT26 cells, using the anti-HiBiT antibody and blotted for ADPr. The new data shown in Supplementary Fig. 3, show enrichment of 75 kDa and 100 kDa ADPr bands with PARP7 and AHR pulldowns respectively.

3. In Fig. 3, the authors state that the "shift" in AHR after AHR agonist treatment results from ADP-ribosylation. While using a PARP7 inhibitor prevents this shift, supporting their claim, it is also possible that the shifted band indicates phosphorylated AHR. In fact, AHR agonists are known to cause AHR phosphorylation and affect its nuclear import (PMID: 37696456). If the authors claim that the shifted band is ADP-ribosylated AHR, they should provide supporting evidence for this assertion. Otherwise, they should temper their claim and consider other PTMs like phosphorylation.

To address this, we have performed treatment of cell lysates with a Glu/Asp/Cys ADPr-specific MacroD hydrolase from *Streptococcus pyogenes* which removes Glu, Asp, Cys ADPr that are amino acids targets for PARP7 (1-3). This resulted in the removal of the TAK243-induced ADPr bands and a downward shift of the AHR band. The new data are show in Supplementary Fig. 4.

1. Lavanya H Palavalli Parsons Sridevi Challa Bryan A Gibson Tulip Nandu MiKayla S Stokes Dan Huang Jayanthi S Lea W Lee Kraus (2021) Identification of PARP-7 substrates reveals a role for MARylation in microtubule control in ovarian cancer cells eLife 10:e60481.

2. Kelsie M Rodriguez Sara C Buch-Larsen Ilsa T Kirby Ivan Rodriguez Siordia David Hutin Marit Rasmussen Denis M Grant Larry L David Jason Matthews Michael L Nielsen Michael S Cohen (2021) Chemical genetics and proteome-wide site mapping reveal cysteine MARylation by PARP-7 on immune-relevant protein targets eLife 10:e60480.

3. Wierbiłowicz K, Yang CS, Almaghasilah A, Wesolowski PA, Pracht P, Dworak NM, Masur J, Wijngaarden S, Filippov DV, Wales DJ, Kelley JB, Ratan A, Paschal BM. Parp7 generates an ADP-ribosyl degron that controls negative feedback of androgen signaling. EMBO J. 2025 Sep;44(17):4720-4744.

4. In Fig. 4A, the controls showing that the siRNAs are functioning are missing. Additionally, the authors should use PARP7i and TNK1/2i to demonstrate that the DTX2 KD-induced ADP-ribosylation of the 75 kDa band is mediated by PARP7 and that the two bands around 150 kDa depend on TNK1/2.

We have performed western blots to show DTX2 and HUWE1 knockdowns. Validated commercial antibodies for the rest of the siRNA targets are not available so we performed qPCR analysis instead. The data are shown in Supplementary Fig. 5. The only siRNA that resulted in a knockdown of less than 80% was that of DTX1. We further validated that the DTX2 knockdown-induced ADP-ribosylation (75 kDa band) belongs to PARP7 as PARP7i treatment removed the signal. The two ADP-ribosylation bands around 150 kDa induced by RNF146 knockdown were eliminated with TNKS1/2i treatment (Supplementary Fig. 6).

5. In Fig 4B, C, DTX2 KD in the presence of the AHR agonist does not appear to rescue AHR levels. Do the authors have an explanation for this? In the presence of the AHR agonist, the authors state that PARP7 quantitatively ADP-ribosylates AHR (Fig. 3). If PARP7-mediated degradation of AHR is dependent on DTX2, should DTX2 knockdown prevent the AHR agonist-mediated decrease in AHR levels? The authors should address this experimentally or in the text.

The tapinarof treatments were performed for 24 h. During this time, other E3 ligases may regulate AHR stability and reduce its levels even in DTX2-depleted cells. To address this, we performed a 2 h. The shorter treatment produced a clearly upregulated AHR levels in DTX2 knockouts compared to wild type cells. This suggests that DTX2 degradation of AHR happens quickly. The new data are presented in Fig. 5B.

6. In Fig. 5, it is unclear whether PARP7i prevents AHR degradation. This could suggest that an AHR agonist induces AHR degradation in a manner that is independent of PARP7. The authors should quantify these results for clarity. Additionally, in the context of DTX2 KO, AHR levels stay lower even when an AHR agonist is present. If DTX2 regulates AHR stability, I expect AHR levels to be higher in this condition. Can the authors explain this?

We have quantified these western blotting results and indeed PARP7i does not prevent AHR degradation. Moreover, AHR agonist tapinarof reduces AHR levels to a similar extent in DTX2 KO compared to wild type A549 cells. In Fig. 5B we now demonstrate that a 2 h AHR agonist treatment results in a substantially increased AHR level in DTX2 KO compared to wild type. In addition, PARP7i prevents AHR degradation with 2 h AHR agonist treatment. We hypothesise that AHR degradation through PARP7-DTX2 happens on more immediate timescales. It is possible that the long-term tapinarof treatment may trigger other E3 ligases and other degradation systems that act independently of the ADPr signal.

7. In Fig. 6F, the authors should determine if PARP7i impacts nuclear AHR levels upon AHR agonist treatment in WT cells.

We have added and quantified the IF images of PARP7i + agonist (tapinarof) in Fig. 6D and 6F. PARP7i indeed increases AHR levels upon agonist treatment compared to the agonist alone (Fig. 6F). This is consistent with PARP7i stabilising AHR by preventing its ADP-ribosylation and subsequent proteasomal targeting by DTX2.

8. In Fig. 6, why do nuclear AHR levels in the presence of AHR agonist decrease even in the presence of TAK243? Is there a change in localization (it's hard to discern from the images).

TAK243 blocks the first enzyme in the ubiquitin pathway but the proteasome would still degrade already ubiquitylated substrates. We have now performed imaging with a confocal microscope to obtain a clearer image. Our new data may suggest that in the TAK243 condition there is more and widespread AHR signal in the cytoplasm (where it undergoes proteasomal degradation) compared to the tapinarof + TAK243 condition (indicated with red outline). Most of the AHR resides in the nucleus where intensity of the AHR staining is the highest (indicated with blue arrows).

9. In Fig. S1, it appears that PARP7i partially restores the TAK243-dependent decrease in total nuclear ubiquitin. This is an interesting result that the authors should further discuss in the text. In any event, the authors should provide a statistical analysis.

We have quantified these IF results and added further hypotheses in the discussion. This is now shown as Supplementary Fig. 9.

10. The authors should specify in the figure legend the specific inhibitors used and their concentrations, as these are included in the methods but are better placed in the legend for easier access by the reader.

We have specified the inhibitor information and concentrations used in figure legends.

11. Page 15, Line 243: "bot" should be "blot"

We have corrected this mistake.

Referee #3:

1. TAK243 was used at 1 μ M according to the methods. Were other concentrations attempted and this found to be optimal? This study will likely serve as a resource for the clever trick of inhibiting ubiquitylation, so it could be helpful to report whether any optimization of conditions was necessary.

We thank the reviewer for highlighting the importance of optimisation. Before proceeding with co-treatment experiments we had performed a titration of TAK243 and showed that 1 μ M of TAK243 is the minimum concentration needed to induce ADPr. These data are now shown in Supplementary Fig. 1. We used 1 μ M of TAK243 for all other experiments in the manuscript.

2. page 6, lines 99,100. Suggest to find a different term than colocalise for the experiments trying to determine whether the 75 kDa band is indeed PARP7. Overlapping migration?

We have changed this to “overlap”.

3. Prior to the multi-color experiments in Figure 4, the shift in AHR migration is not always noticeable in the gel images. Is there a way to highlight the difference with gel markings?

We have made this band shift clearer in Supplementary Figure 4.

4. The discussion and some of the references refer to an ADP-ribose-ubiquitin dual modification. Is there any evidence that this is the modification that arises from PARP7 and DTX2? It would be useful to have at least some discussion on this point.

We have performed DTX2-mediated ubiquitination of ADP-ribosylated AHR (Supplementary Figure 7A) and a PARP7-derived ADPr peptide as this point was also raised by reviewer 1.

Dr. Ivan Ahel
University of Oxford
Sir William Dunn School of Pathology
South Parks Road
Oxford OX1 3RE
United Kingdom

27th Oct 2025

Re: EMBOJ-2025-121671R
Ubiquitin pathway blockade reveals endogenous ADP-ribosylation marking PARP7 and AHR for degradation

Dear Ivan,

Thank you for submitting your revised manuscript to The EMBO Journal. Two of the original referees have now assessed it once more, and I am happy to say they both were fully satisfied with the revisions. After incorporation of the following remaining editorial issues, we should therefore be able to proceed with formal acceptance of the study:

- Please submit the main text as a text-only file without inclusion of figures, and all main Figures as individual, image-only files with sufficient resolution/quality for production. Ideally, all figures should be uploaded in TIFF format, currently only Figure 6 is provided at acceptable resolution for publication.
- Please adjust the order and the headers of the different manuscript sections: Title page with complete author information, Abstract, Keywords, Introduction, Results, Discussion, Methods, Data Availability, Acknowledgements, Disclosure and Competing Interests Statement, References, Main Figure Legends, Tables.
- On the abstract page of the manuscript, please include 4-5 general keyword terms to enhance searchability.
- Please adjust the format of the reference list and of the in-text citations according to EMBO Journal format (alphabetical order, author name et al + year...). Also, please make sure that each reference is complete with citation year, volume, and page/locator numbers (currently missing in several of them), and that alternative DOI information is only included for pre-publication manuscripts that do not have a formal citation description yet. Finally, please adjust the format for citation of bioRxiv preprints as specified in our author guidelines. The citation in the text should be: "(PREPRINT: NAME1 et al, YEAR)"; in the reference list: "Author NAME1, Author NAME2, ... (YEAR) article title. bioRxiv doi: XXX"
- As we are switching from a free-text author contribution statement towards a more formal statement based on Contributor Role Taxonomy (CRediT) terms, please remove the present Author Contribution section and instead specify each author's contribution(s) directly in the Author Information page of our submission system during upload of the final manuscript. See <https://casrai.org/credit/> for more information.
- Please update the Data Availability section with a permanent access link to the deposited BioImage Archive datasets, rather than the temporary referee access link.
- Please specify the exact p-values in the legends of figures 6B, C, E, F, H, I
- Finally, please also double-check the resolution of the Appendix figures, making sure that compiling them in the Appendix PDF has not compromised them.

I am returning the manuscript to you for a final round of minor revision, solely to allow you to make these modifications and upload the revised files. Once we will have received them, we should be ready to proceed with formal acceptance and production of the manuscript.

With kind regards,

Hartmut

*** PLEASE NOTE: All revised manuscripts are subject to initial checks for completeness and adherence to our formatting guidelines. Revisions may be returned to the authors and delayed in their editorial re-evaluation if they fail to comply to the following requirements (see also our Guide to Authors for further information):

9) To facilitate reproducibility and cross-laboratory adoption of methodologies, please structure the Materials & Methods section as outlined in our guide to authors, including a completed Reagents and Tools Table that can be downloaded from our author guidelines as well (<https://www.embopress.org/page/journal/14602075/authorguide#structuredmethods>).

10) Digital image enhancement is acceptable practice, as long as it accurately represents the original data and conforms to community standards. If a figure has been subjected to significant electronic manipulation, this must be clearly noted in the figure legend and/or the 'Materials and Methods' section. The editors reserve the right to request original versions of figures and the original images that were used to assemble the figure. Finally, we generally encourage uploading of numerical as well as gel/blot image source data; for details see: embopress.org/page/journal/14602075/authorguide#sourcedata

Further information is available in our Guide For Authors:

In the interest of ensuring the conceptual advance provided by the work, we recommend submitting a revision within 3 months (25th Jan 2026). Please discuss the revision progress ahead of this time with the editor if you require more time to complete the revisions. Use the link below to submit your revision:

Link Not Available

Referee #1:

This manuscript builds upon previous work indicating PARP7 regulates AHR stability following AHR activation. The use of TAK243 to increase levels of endogenous ADP-ribosylation is interesting and novel. Given that AHR and PARP7 are targets for cancer therapy, the findings here may be useful for future translational studies.

The original version of this paper was interesting and the data were strong. The previous strengths remain in the revised version:

- The main conclusions of the work are supported by several experiments across multiple cell lines.
- The manuscript is well written and the data are presented clearly.
- The discussion section does a good job of placing these observations in the context of previous literature and provides interesting speculations about potential future directions.

The revised manuscript has addressed concerns about

- the need for additional characterization of the post-translational modification(s) mediated by PARP7 and DTX2 on AHR.
- limitations of immunoassays.

This is a nice paper that extends previous observations from other labs that, quite frankly, missed the mark.

Referee #2:

The authors have sufficiently addressed all of my comments. Additionally, the DTX2 KO data is a nice addition.